


Planktic foraminifera and structure of surface water masses at the SW Svalbard
margin in relation to climate changes during the last 2000 years

Katarzyna Zamelczyk[1], Tine Lander Rasmussen[1], Markus Raitzsch[2], Melissa Chierici[3]

[1] Centre for Arctic Gas Hydrate, Environment and Climate, Department of Geoscience, UiT -
The Arctic University of Norway, N-9037 Tromsø, Norway

[2] MARUM - Zentrum für Marine Umweltwissenschaften, Universität Bremen, Leobener Str.,
28359 Bremen, Germany

[3] Institute of Marine Research, Box 6404, 9294 Tromsø, Norway

*Correspondence to:* Katarzyna Zamelczyk (katarzyna.zamelczyk@uit.no)

**Abstract.** We present a high-resolution record of properties in the subsurface (250–100 m),
near surface (100–30 m) and surface (30–0 m) water masses at the SW Svalbard margin in
relation to climate changes during the last 2000 years. The study is based on planktic
foraminiferal proxies including the distribution patterns of planktic foraminiferal faunas, $\delta^{18}O$

5    and $\delta^{13}C$ values measured on *Neogloboquadrina pachyderma*, *Turborotalita quinqueloba,* and
*Globigerinita uvula*, Mg/Ca- , $\delta^{18}O$- and transfer function-based sea surface temperatures,
mean shell weights and other geochemical and sedimentological data. We compared paleo-
data with modern planktic foraminiferal fauna distributions and the carbonate chemistry of the
surface ocean. The results showed that cold sea surface conditions prevailed at ~ 400–800 AD

10    and ~ 1400–1950 AD are associated with the local expression of the Dark Ages Cold Period
and Little Ice Age, respectively. Warm sea surface conditions occurred at ~ 21–400 AD, ~
800–1400 AD and from ~ 1950 AD until present and are linked to the second half of the
Roman Warm Period, Medieval Warm Period and recent warming, respectively. On the
centennial to multi-centennial time scale, sea surface conditions seem to be governed by the

15    inflow of Atlantic water masses (subsurface and surface) and the presence of sea-ice and the
variability of sea-ice margin (near surface water masses). However, the close correlation of
sea surface temperature recorded by planktic foraminifera with total solar irradiance (TSI)
implies that solar activity could have exerted a dominant influence on the sea surface
conditions on the decadal to multidecadal time scale.

**1 Introduction**



The inflow of warm Atlantic Water via the West Spitsbergen Current (WSC) to the
Arctic is considered the main control of climate in the Fram Strait over millennial (e.g.,
Dokken and Hald, 1996; Rasmussen et al., 2007; Müller et al., 2009; Jessen et al., 2010;
Zamelczyk et al., 2012; 2014; Telesinski et al. 2017) to centennial timescales (e.g. Sutton and
Hodson, 2005; Majewski et al. 2009; Spielhagen et al., 2011; Zamelczyk et al., 2013; Werner
et al., 2016; Pawłowska et al. 2016). Atlantic water masses transported to the north via the
WSC converts warm and saline surface water to cold and less saline deep water masses. The
heat loss and decrease in salinity depend on the interactions of the surface water with cold
melt water and cold air masses (Brambilla et al., 2008). Millennial scale variability in surface
ocean circulation and properties of the Atlantic water masses have been documented for the
Holocene interglacial period (e.g., Bianchi and McCave, 1999; Oppo et al., 2003; Giraudeau
et al., 2010; Van Nieuwenhove et al., 2016) and of past glacial periods (e.g., Hoff et al.,
2016). However, apart from a few studies west of Svalbard (Majewski et al. 2009; Spielhagen
et al. 2011; Werner et al., 2011; Zamelczyk et al., 2013; Pawłowska et al. 2016), little is still
known about the natural variability of surface and subsurface ocean circulation over the
western Svalbard margin on decadal to centennial timescales beyond the last ~ 200 years. The
Roman Warm Period (RWP, ~ 200 BC–400 AD), the Dark Ages (DA, ~ 400–800 AD), the
Medieval Warm Period (MWP, ~ 800–1400 AD) and the Little Ice Age (LIA, ~ 1400–1950
AD (Grove, 2004)) are well-documented in social history of the North Atlantic region (e.g.
Lamb 1977). Prior to the anthropogenic changes during the last century, interactions between
the warm Atlantic Water and supply of cold polar water were controlled by natural climate
anomalies of the past two millennia (e.g., Werner et al., 2011; Zamelczyk et al., 2013;
Pawłowska et al. 2016). Freshwater input from melting sea-ice and high frequency physical
and chemical changes occurring in the upper surface ocean are relevant for defining and
assessing the dynamics of the present ocean and climate changes. These dynamics are an
essential part of the deep-water production and hence the Atlantic Ocean's thermohaline
circulation (e.g., Rudels, 1995). However, existing Arctic instrumental records are sparse and
burdened with great uncertainties with regard to the significance of the recent warming,
natural variability and the superimposed anthropogenic influence. Thus, more studies are
required for a better understanding of the timing, spatial extent, local appearance and the
magnitude of changes during the past two millennia compared to modern instrumental



variability indicating the warming of the last century, particularly in relation to changes in the flow of Atlantic Water.

Planktic foraminifera are a powerful tool to reconstruct sea surface water mass conditions of the past. Individual species of planktic foraminifera preferentially live in

specific water masses at and near the sea surface and can be used to assess the properties and variability of the upper water column. The ecologically distinctive species in the polar modern planktic foraminiferal assemblages are *Turborotalita quinqueloba, Neogloboquadrina pachyderma* (Carstens et al., 1997; Volkmann, 2000), and recently also, *Globigerinita uvula* and its variant *Globigerinita uvula minuta* (Hulot, 2015; Meilland et al., 2015; 2016; Schiebel

et al., 2017).

The purpose of the present study is to reconstruct the properties of the upper ~ 250 m of the water column over the southwestern Svalbard margin for the past 2000 years to get a better understanding of the flow of Atlantic Water in relation to climate over historical times on centennial to decadal time scales. The sediment core has been retrieved from Storfjorden

Fan south-west of Svalbard in the Atlantic Water influenced part of the Fram Strait. The study is based on the distribution patterns and concentration of planktic foraminifera, and sea surface temperatures (SST) based on Mg/Ca ratio, $\delta^{18}O$ values measured on *N. pachyderma* and *T. quinqueloba*, and transfer function-based SST. In addition, we combine the paleo-record with modern planktic foraminifera plankton tows and the carbonate chemistry of the

surface ocean and apply interspecific isotopic relationships and the main, species-specific, depth habitats of *G. uvula, T. quinqueloba* and *N. pachyderma* to infer oceanic conditions of the surface, near surface and subsurface water masses, respectively. Furthermore, based on the main depth habitats of *N. pachyderma, T. quinqueloba* and *G. uvula*, we use the difference in the $\delta^{18}O$ values between *N. pachyderma* and *T. quinqueloba* ($\Delta\delta^{18}O_{Np-Tq}$) as an indicator of

the subsurface-to-near surface Atlantic Water relative inflow and between *T. quinqueloba* and *G. uvula* ($\Delta\delta^{18}O_{Tq-Gu}$) as an indicator of relative changes in freshening and stratification of the surface waters in the past. For supporting information, we also use dissolution proxies and other geochemical and sedimentological data. We compare our results to other studies of the Late Holocene from the area to better assess the nature of large-scale climate forcings.

## 1.1 Oceanographic setting



The near surface oceanographic setting SW off Svalbard Archipelago is characterized by the interaction between temperate and saline Atlantic Water (S>34.92; T>0 °C) transported by the West Spitsbergen Current (WSC) from south and cooler and less saline Arctic water masses conveyed by the East Spitsbergen Current (ESC) from the east (Fig. 1). The Coastal

Current (CC), which is the extension of the ESC, contains a mixture of Arctic Water and sea-ice transported from the northern Barents Sea and/or produced in Storfjorden. The two contrasting water masses generate an oceanic front, the Arctic Front (AF) that is variably located near the coring site.

**2 Material and methods**

**2.1 Modern oceanographic data and modern planktic foraminifera**

Temperature and salinity were measured via a SBE Seabird 911 plus CTD

(conductivity-temperature-depth) rosette equipped with Niskin bottles aboard the RV Helmer Hanssen at the study site in October 2012, July 2014 and April 2015.

In April 2015, in addition to the CTD record, water samples were collected for the study of the surface to subsurface ocean carbonate chemistry (Fig. 2I). The carbonate chemistry was determined using measurements of total dissolved inorganic carbon ($C_T$) and

total alkalinity ($A_T$,) on samples collected at 200, 100, 50 and 5 m water depth and analyzed at the Institute of Marine Research, Tromsø, Norway. $C_T$ was determined by using coulometric titration on a VINDTA system and $A_T$ by using potentiometric titration with weak HCl on a VINDTA system (Marianda, Germany). The analytical methods and sampling for $C_T$ and $A_T$ determination have been described in detail in Dickson et al. (2007). In 2015, the precision

were ± 1 µmol kg$^{-1}$ for both $A_T$ and for $C_T$. The accuracy of $A_T$ and $C_T$ were determined using Certified Reference Material (CRM) supplied by A. Dickson (San Diego, USA) by applying a correction factor to the measured values based on the measured CRM value. Values of $C_T$, $A_T$ and salinity, temperature, and depth for each sample were used as input parameters in a $CO_2$-chemical speciation model ($CO_2$SYS program) (Pierrot et al., 2006) to calculate the calcium

carbonate saturation state ($\Omega_{Ca}$), pH and the carbonate-ion concentration [$CO_3^{2-}$]. The calculations were performed on the total hydrogen ion scale using the hydrogen sulphate [$HSO_4^-$] dissociation constant of Dickson (1990). We used the carbonate system dissolution constants from Mehrbach et al. (1973) refit by Dickson and Millero (1987). The concentration



of calcium, $[Ca^{2+}]$ is assumed to be proportional to the salinity according to the equation $10.28 \times S/35 \mu mol \ kg^{-1}$ (Mucci, 1983). The thermodynamic solubility products for calcite (Ksp) are from Mucci (1983).

Living planktic foraminifera were collected immediately after recovering the CTD casts in October 2012 and July 2014 at the coring site using a WP2 (Working Party 2) plankton tow with 90 µm mesh-size net and 0.255 m² opening. In 2012, vertical depth-sampling intervals were 0–50 and 50–200 m, and in 2014, vertical depth-sampling intervals were 0–50, 50–100 and 100–200 m (Fig. 2II). The depth-sampling intervals were assigned based on the distribution of water masses recorded by the CTD. Vertical towing speed was ~

0.5 m s $^{-1}$. Immediately, after recovering the samples were sieved over sieves of mesh-sizes 1000 µm and 63 µm. Fraction > 63 µm was fixed with 98 % ethanol buffered with disodium hydrogen phosphate to prevent dissolution. Additionally, the samples were stained with Rose Bengal (1.0 g l $^{-1}$) in order to discriminate ''living'' foraminifera (cytoplasm-full shell) from ''dead/transported'' (cytoplasm-empty) individuals (Lutze and Altenbach, 1991). Samples

were kept cool until analysis performed in the laboratory at UiT, the Arctic University of Norway, Tromsø. Only living specimens were considered for analysis. The living planktic foraminifera were counted and concentrations calculated as number of specimens per cubic meter (ind. m⁻³).

**2.2 Sediment core**

Box core HH12-1206BC (76°24' N; 012°58' E) was retrieved from Storfjorden Fan from 1520 m water depth during a cruise with RV *Helmer Hanssen* to the Fram Strait in October 2012 (Fig. 1). A tube of inner diameter 10 cm was pushed into the sediment for

subsampling. The 30.5 cm long sediment core was sampled at 0.5 cm intervals in the laboratory at UiT, the Arctic University of Norway, Tromsø. Subsequently, the samples were weighed, freeze-dried, weighed again, and the water content (%) was calculated. The dry samples were wet sieved through mesh-sizes 1 mm, 500 µm, 100 µm, and 63 µm following the preparation methods of Feyling-Hanssen et al. (1971) and Knudsen (1998). The residues

were dried at room temperature and weighed.

The grain size distribution was calculated as the weight percentage of the residue relative to total dry weight of the sample. The ice-rafted debris (IRD) was counted in the >500





and >150 µm size-fractions. The 150 µm size-fraction was obtained by dry-sieving of the 100–500 µm fraction after counts and identification of foraminifera.

Total carbon (TC) and total organic carbon (TOC) were measured in bulk samples using a Leco CS-200 induction furnace instrument. The weight percentage (wt %) of TC and
TOC was calculated in intervals of 1–2 cm. The $CaCO_3$ content (wt %) was calculated using the following equation: $CaCO_3 = (TC-TOC) \times 8.33$ (Espitalié et al., 1977).

The fraction 100–500 µm was split and a representative number of planktic foraminifera specimens (>300) were picked from the >100 µm size fraction, counted and identified to species level. The number of foraminifera per gram dry weight sediment was
calculated.

Relative changes in mean foraminiferal shell weight can be interpreted as changes in shell thickness, changes in preservation and the extent of calcification (Lohmann, 1995; Barker and Elderfield, 2002; Barker et al., 2004; Zamelczyk et al., 2012, 2013). Therefore, shells of planktic foraminiferal species *N. pachyderma* and *T. quinqueloba* were weighed
using a Sartorius microbalance (model M2P, 0.1 µg sensitivity). The shells were picked from two narrow size ranges of 100–125 and 150–180 µm. The small- and large-size shells represent different life stages, the juvenile and adult forms, respectively. Mean shell weights were calculated by dividing the weighed total mass of 20–30 individuals in each size range by the number of weighed shells. Due to low abundances of *T. quinqueloba*, 16 samples from
100–125 µm size-fraction and 15 weighed samples from 150–180 µm size-fraction the total mass contained less than 10 individuals. All measurements were repeated 3 times. Shells with sediment fill or showing signs of mechanical corrosion at the outer shell surface or having visually detectable secondary calcite crusts were omitted.

Shell fragments were counted as a measure of dissolution and an indicator for the
general preservation state of specimens and species (Berger et al., 1970; 1982). The number of fragments per gram dry sediment was calculated and the percent fragmentation was calculated relative to the total numbers of planktic foraminifera g $^{-1}$ and the total number of fragments g $^{-1}$ in a sample.

Stable isotopes were measured on the three species *N. pachyderma*, *T. quinqueloba*
and *G. uvula minuta* from size-fractions: 125–150 µm, 150–180 µm and 100–150 µm, respectively. The measurements were done at the Leibniz Laboratory for Radiometric Dating and Stable Isotope Research in Kiel, Germany, using an automated Carbo-Kiel device connected to a Finnigan MAT 253 and MAT 252 mass spectrometers. Results refer to the

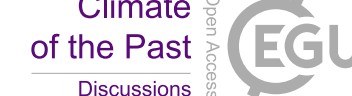

Vienna Pee Dee Belemnite (VPDB) standard. The external analytical reproducibility was <0.06 ‰ and <0.03 ‰ for $\delta^{18}O$ and $\delta^{13}C$, respectively. Measurements on all three species were carried out at 0.5 to 2 cm intervals. Between 7 and 10 cm core depth, absence of *T. quinqueloba* and *G. uvula* specimens prevented analysis.

5        The first top 12 samples (upper 6.5 cm) were analyzed for activity profiles of $^{210}Pb$ and $^{137}Cs$ in order to determine the modern sedimentation rate, assess the recovery of core top sediments and to establish the age of the youngest sediments. Freeze-dried and homogenized samples were analyzed using a high-resolution germanium diode gamma detector and multichannel analyzer gamma counter at the Centre d'études Nordiques (CEN), Université

Laval (Canada). Prior to analysis, the samples were put into plastic vials to rest for at least three weeks to attain the secular equilibrium. The linear apparent sedimentation rate was calculated from the decrease of excess $^{210}Pb$ activities with sediment depth following McKee et al. (1983). Excess $^{210}Pb$ activities were determined by subtracting the average supported activity (average of $^{214}Pb$, $^{214}Bi$ and $^{226}Ra$) from the total activity (Fig. 3a). The $^{210}Pb$-derived

sedimentation rate was confirmed using the first occurrence of $^{137}Cs$ as a marker of the early 1950s (Robbins and Edgington, 1975).

       Nine accelerator mass spectrometry (AMS) $^{14}C$ ages were achieved on monospecific samples of *N. pachyderma*, except ages at 4.5–6.5 and 13–14 cm, where all available planktic foraminifera were picked. Measurements were performed at the Poznań Radiocarbon

Laboratory, Poland (Table 1). The dates were calibrated into calendar ages using the calibration program CALIB Rev. 7.0.2 (Stuiver & Reimer, 1993) and the Marine13 calibration data set (Reimer et al., 2013). A global average marine reservoir age of 400 radiocarbon years and $7 \pm 11$ $^{14}C$ years of local deviation ($\Delta R$) (Mangerud et al., 2006) was applied. The sample from 2.0–3.5 cm core depth most likely contained modern, post-A-bomb

carbon, indicating a post-1950 age. Samples at 4.5–6.5 cm and at 9.5–11.5 cm gave inverted ages probably due to mixing of sediment either representing redeposition events or a possible error introduced by the very small amount of material (0.01–0.02 mgC) used for measurements of these samples (Table 1). These three dates ware not used in the construction of the age model. The calibrated ages are reported as years AD with a $2\sigma$ standard error age

range.

       Trace element ratios of Mg/Ca and Al/Ca were measured on *N. pachyderma* (60–130 shells/sample) at 0.5 to 2 cm resolution. Clean, well-preserved shells with no visible signs of dissolution were picked from a narrow size fraction of 150–180 μm. Prior to analysis, the





foraminiferal shells were crushed and cleaned in the clean-lab at the Alfred Wegener Institute (AWI), Bremerhaven, Germany. The cleaning procedure principally followed the protocol given in Barker et al. (2003). The procedure included four steps: clay removal with boron-free MQ water and methanol, oxidation of organic matter in NaOH-buffered $H_2O_2$, weak acid

leach in 0.001 N HCl, and carbonate dissolution in 0.2 N HCl. Subsequently, from each sample an aliquot of 5 µL was diluted with 2 % $HNO_3$ to determine the Ca concentration. Finally, the remaining residue was diluted with 2 % $HNO_3$ to obtain a Ca concentration of 15 to 20 ppm. The samples were analyzed using a Nu AttoM high resolution double-focusing inductively coupled plasma mass spectrometer at AWI, Bremerhaven. Five replicates were

carried out for every sample. Long-term precision ($2\sigma$) of analyses for Mg/Ca is 2.8 %. Since sample material was very limited, analyses were conducted at a Ca concentration of ~ 15 ppm. The measurement uncertainty was determined by analyzing the carbonate standard JCt-1 (giant clam) yielding Mg/Ca ratios of $1.28 \pm 0.05$ ($\pm 2\sigma$) mmol/mol, which is within 1 % of the values reported in the literature (Inoue et al., 2004; Hathorne et al., 2013).

Al/Ca ratios in the foraminiferal samples were used to detect potential clay contamination that might bias the Mg/Ca ratios measured in the foraminiferal calcite (Rosenthal et al., 2000; Barker et al., 2003). No significant correlation between Al/Ca and Mg/Ca ($R^2 = 0.15$) was found. Three samples with Al/Ca >0.80 mmol/mol (Barker et al., 2003) were omitted. No correlation between high Al/Ca ratios and outliers within the Mg/Ca

data set was present. We conclude that the measured high Al/Ca ratios did not affect the Mg/Ca signal. In the results and discussion sections, we will discuss the Mg/Ca-based temperature profiles instead of the Mg/Ca ratios since both are exactly the same and the temperature reconstructions allow us to compare with other temperature reconstructions in our study.

### 2.2.1 SST calculations based on $\delta^{18}$O, Mg/Ca ratios and transfer function

Reconstructions of sea surface temperatures (SST) were calculated based on measured $\delta^{18}$O, Mg/Ca ratios and transfer functions. The average calcification temperature, hence sea

surface temperatures were calculated from measured $\delta^{18}$O values ($SST_{\delta^{18}O}$) using the equation of Shackleton (1974):

(Eq.1) T (°C) = 16.9 − 4*($\delta^{18}O_{\text{foram, vs. V-PDB}}$ + $\delta^{18}O_{\text{vital effect, vs. V-PDB}}$ − $\delta^{18}O_{\text{water, vs. V-SMOW}}$),



where $\delta^{18}O_{water}$ is standard mean ocean water composition (V-SMOW). Conversion from the Standard Mean Ocean Water (SMOW) scale to calcite on the Pee Dee Belemnite (PBD) scale was done by subtracting 0.2 ‰ (Shackleton, 1974). For tentative temperature calculations, a

constant value of 0.3 ‰ PDB for paleo-$\delta^{18}O_{water}$ was used. This value is considered as the average modern value on the West Spitsbergen margin at 25–500 m water depth (Meredith et al., 2001). A vital effect correction of 0 and +0.7 ‰ was applied for *N. pachyderma* and *T. quinqueloba*, respectively (Jonkers et al., 2010). $SST_{\delta^{18}O}$ are calculated assuming a constant salinity of 35.1.

The measured Mg/Ca ratios were used to calculate temperatures ($SST_{Mg/Ca}$) by using the species-specific (*N. pachyderma*) linear equation of Kozdon et al. (2009):

(Eq. 2) Mg/Ca (mmol/mol) = 0.13*T + 0.35

This calibration is based on core-top samples of *N. pachyderma* from the Nordic Seas and produces reliable $SST_{Mg/Ca}$ estimates at temperatures above ~ 3 °C (Kozdon et al., 2009).

Transfer functions were used to reconstruct SST based on the census data of the planktic foraminiferal species distribution. The statistical method is built on a comparison between modern oceanographic data (summer SST at 10 m depth) and compositions of the

planktic foraminiferal faunas in the >100 μm size-fraction from surface sediments collected in the Arctic areas influenced by Atlantic Water in the Nordic Seas (Husum and Hald, 2012). The data were processed using the data analysis program C2 version 1.3 (Juggins, 2010). Several statistical methods were tested. The most precise estimate assessed by the root mean squared error of prediction (RMSEP) is indicative for the most predictive transfer function

model, but it may also carry a possible spatial auto-correlation bias in the training set causing overoptimistic estimates of the RMSEP (Telford and Birks, 2005). This bias is most reduced for the Maximum Likelihood (ML) model (Telford and Birks, 2005). The ML based SST reconstruction is therefore considered as the best representative model for the present record. Moreover, ML-calculated SSTs are close to modern water temperature ranges of the core site

and follow the actual fluctuations in species composition of the planktic foraminiferal faunas.

**3 Results**



### 3.1 Modern data

### 3.1.1 Temperature and salinity

The CTD casts identified a ~ 50 m thick, mixed layer of melt water (MMW) characterized by low salinity (except October 2012) and relatively high temperatures (T ~ 6.3° C in October, 7.9–5.4 °C in July and 4.1–4.7 °C in April) (Fig. 2Ia,b). From ~ 50 m to about 500–600 m water depth a thick layer of warm and saline Atlantic Water is defined by temperatures ranging from 6.3 °C to 1.5 °C in October 2012, 5.4 °C to 0.6 °C in July 2014,

and from 4.7 °C to 2.2 °C in April 2015 (Fig. 2Ia,b). The Atlantic water masses overly the cold intermediate waters (T< 0 °C) generated by convection in the Nordic Seas (e.g., Blindheim et al., 2000). In summer and autumn, the coring site is ice-free, whereas in spring and winter the sea-ice incursion depends on the intensity of drift ice carried by the ESC and the production of sea-ice in Storfjorden (NSIDC).

### 3.1.2 Carbonate chemistry of the study area

       The $[CO_3^{2-}]$ and $\Omega_{Ca}$ are key variables to monitor and assess a change in ocean carbonate chemistry and give information on the thermodynamic potential to dissolve $CaCO_3$.

The $\Omega_{Ca}$ value is expressed by the product of concentrations of calcium ions $[Ca^{2+}]$ and $[CO_3^{2-}]$ in sea water divided by the solubility product (Ksp) at a given temperature, salinity and pressure. When $\Omega >1$, $CaCO_3$ will be kept in solid state and when $\Omega <1$, $CaCO_3$ will tend to dissolve. In April 2015, the surface and subsurface waters (upper 200 m) were well saturated with respect to calcite ($\Omega_{Ca}$ ranges from 2.8 at the surface to 2.66 near 200 m water depth, Fig.

2Ic). Both, the carbonate ion concentration $[CO_3^{2-}]$ and the concentration of $CO_2$ (shown as pH), showed a decrease from the surface to 200 m water depth (Fig. 2Ic).

### 3.1.3 Modern distribution and ecology of living high-latitude planktic foraminifera

As our plankton tows data are collected in two different years and two different seasons we cannot cover inter-annual and inter-seasonal variability of planktic species composition and their main living depth. We therefore support our data with published results



to establish a record of the modern distribution of planktic foraminiferal species in the surface
waters of the study area.

At our study site, maximum absolute abundances of *T. quinqueloba* are found in July
2014 at 0–100 m water depth (Fig. 2II). *Turborotalita quinqueloba* is generally considered a
photic-zone species due to its symbionts and food requirements (Hemleben et al., 1989). In
open and ice-free waters of the Fram Strait, the photic zone and the calcification depth of *T.
quinqueloba* can reach ~ 200 m water depth (Carstens et al. 1997). However, during
spring/summer stratification, peaks in primary production absorb and reflect most of the
sunlight limiting the depth of the photic zone and cause upward migration of the planktic
foraminiferal species (Hemleben et al., 1989). In the Fram Strait, Simstich et al. (2003) and
Sarnthein and Werner (2017) report an apparent calcification depth of *T. quinqueloba* not
deeper than 30–70 m.

In both seasons, *N. pachyderma* showed a maximum in absolute abundances in the
deeper layers of the surface ocean, where Atlantic water masses occupy the water column
throughout the year (150–200 m water depth, Fig. 2Ia,b). Generally, this species has been
classified as deep-dwelling species, which grows and calcifies around 200 m water depth
(Volkmann et al., 2001; Simstich et al., 2003; Sarnthein and Werner, 2017). However, the
depth of maximum abundance varies from region to region suggesting that their abundance
and shell chemistry are tied to density horizons or other conditions (such as food availability)
that become more clearly defined with depth (e.g., Volkmann et al., 2001; Simstich et al.,
2003; Sarnthein and Werner, 2017). In the Fram Strait, the average depth of calcification of
*N. pachyderma* lies between 100 and 200 m water depth (Bauch et al., 1997). The depth may
vary depending on the water mass distribution and sea-ice cover (e.g., Volkmann et al., 2001).
*Neogloboquadrina pachyderma* seems to descend down to 250 m water depth in the Atlantic
Water off Norway (Simstich et al., 2003) and below 200 m water depth into the warm and
saline WSC identified in the eastern Fram Strait as a subsurface water mass (e.g., Volkmann
et al., 2001; Pados and Spielhagen, 2014; Pados, et al., 2015).

In our plankton tow data from October 2012 and July 2014, *G. uvula* was found at 0–
50 m and 100–200 m water depth in July 2014 (Fig. 2II), which could indicate that this depth
30  is the calcification depth of the species (Fig. 2II). However, its presence in the upper 50 m
suggest that it rather calcifies in the mixed uppermost layer of the ocean and probably
migrated to the deeper water masses at a later stage. This assertion would be consistent with
the average living depth of *G. uvula minuta* in the subtropical North Atlantic, which has been



documented to be at 15 m water depth with average dispersion of ± 9 m (Rebotim et al., 2017). Generally, it has been observed that *G. uvula* lives in the upper 30–50 m water depth (Boltovskoy et al., 1996, H. Bauch, unpublished data) and is considered very tolerant of hyposaline conditions (Hemleben et al., 1989). This species is capable of living in salinities of

30.5–31, which appears to be the minimum salinity for planktic foraminifera (Boltovskoy and Lena, 1970b).

### 3.2 Paleo-data from sediment core HH12-1206BC

### 3.2.1 Age model, sedimentation rate and sedimentology

The $^{210}$Pb and $^{137}$Cs activity profiles show a typical decrease with depth to 6.25 cm. The 3 cm thick surface layer has the highest and near-uniform activities and is interpreted as a surface mixed layer. The presence of this layer implies recovery of the upper surface sediments of the core. From 3.25 cm, $^{210}$Pb activities decrease exponentially down to 6.25 cm

of the core. The average apparent sedimentation rate is 0.08 cm yr$^{-1}$ (Fig. 3a). The $^{137}$Cs activity does not reveal any evident $^{137}$Cs peaks (Fig. 3a). The maximum sedimentation rate is 0.1 cm yr$^{-1}$. However, after taking into account the ~ 3 cm thick surface mixed layer, the sedimentation rate is 0.07 cm yr$^{-1}$ (Fig. 3a), which is close to the $^{210}$Pb-derived value.

The age model was constrained using six of nine AMS $^{14}$C ages (Table 1, Fig. 3b). A constant sedimentation rate between the dated levels was assumed. The age model indicates that the core covers the time period from ~ 21 to ~ 2012 AD (Fig. 3b). The mean sedimentation rate is ~ 16 cm kyr$^{-1}$. Between ~ 1420 and ~ 1060 AD, sedimentation rate is lowest ~ 6 cm kyr$^{-1}$, and in the upper part (~ 1960 AD to present), the sedimentation rate is

highest reaching ~ 53 cm kyr$^{-1}$. This value is considered to be close to value of 80 cm kyr$^{-1}$ estimated by $^{210}$Pb activity profile and to value of 70 cm kyr$^{-1}$ estimated by $^{137}$Cs activity. Due to reversed $^{14}$C dates within the upper ~ 11.5 cm, the time period from ~ 1800 AD to the present is considered to contain potential redeposition events and is therefore interpreted with great caution.

The sediment is dominated by clay and silt (61–96 %) with generally low sand content but a small increase toward the core top (Fig. 4b). The concentration of IRD >1 mm and 500–1000 µm is generally low, however, from ~ 1750 AD to present, the IRD 500–1000 µm increase towards the top (Fig. 4a).



### 3.2.2 Planktic foraminifera, mean shell weights and fragmentation

Generally, the planktic foraminiferal assemblages in core HH12-1206BC are
dominated by *N. pachyderma* (33–88 %) except for the core top where *T. quinqueloba*
dominates (9–43 %) (Fig. 4g,h). The third most abundant species is *Globigerinita uvula/
uvula minuta* (2–20 %) that, until recently, has not been abundantly found and studied in the
Arctic region (Fig. 4i). However, studies of plankton tows collected in 2014 alongside
western Barents Sea have shown that *G. uvula* constituted 25 to 64 % of the planktic
foraminiferal assemblages (Hulot et al., 2015). The increasing abundances of this species in
the living planktic foraminiferal fauna in the Fram Strait (Schiebel et al., 2017) may indicate
the beginning of the long-lasting change of species composition due to changing hydrographic
conditions in the Fram Strait (Walczowski et al., 2017). Therefore, we include visual
documentation and a short presentation of these species. *Globigerinita uvula minuta* is a
morphologically similar to *Globigerinita uvula*, but has larger, more strongly inflated
chambers (Fig. 5). *Globigerinita uvula minuta* is associated with warmer waters (Parker 1962;
Rögl and Bolli, 1973), whereas *Globigerinita uvula* is indicative for relatively cold nutrient-
rich waters and productive zones of the oceanic fronts (Saito et al., 1981; Boltovskoy et al.,
1996; Bergami et al., 2009). Both forms, *Globigerinita uvula* and *Globigerinita uvula minuta*
were found in our record, but not separated. Therefore, we refer to both forms as
*Globigerinita uvula*.

Other species present in the record are *Neogloboquadrina incompta*, *Globigerinita
glutinata*, *Globigerina bulloides* (the two latter are not included in the figure as they were
always <1 %).

Mean shell-weights, of both juvenile (100–125 µm) and adults (150–180 µm) forms of
*N. pachyderma* and *T. quinqueloba* show relatively low variability (Fig. 6a). The shell
weights of both species indicate a similar response to environmental changes and/or
dissolution of the two species (Fig. 6a). The % fragmentation shows an opposite pattern to the
mean shell-weight records (Fig. 6a,b), i.e. high percentage of fragmentation corresponds to
low shell-weight of planktic foraminifera (Figs. 4f, 6b). Increased shell fragmentation and %
of TOC, as well as low % of calcium carbonate in the sediment can be used as evidence for
dissolution due to diagenetic processes (Zamelczyk et al., 2013 and references therein). In
core HH12-1206BC, the %TOC and %$CaCO_3$ is almost constant through the entire record.



Therefore we assume that there are no significant changes in preservation of the planktic foraminifera in the recorded time interval (Fig. 4c,e).

### 3.2.3 Stable isotopes

From ~ 21 AD to ~ 1800 AD, the $\delta^{18}O$ values of *N. pachyderma* generally show an increasing trend except for high values at ~ 380 and ~ 1500 AD, and low values at ~ 450–700 and ~ 1650 AD (Fig. 7Ia). The $\delta^{18}O$ values of *T. quinqueloba* indicate a similar trend, but more variable (Fig. 7Ib). High $\delta^{18}O$ values of *T. quinqueloba* are recorded at ~ 380, ~ 700–

800, ~ 1150, ~ 1380 and ~ 1800 AD, and low $\delta^{18}O$ values are recorded at ~ 130, ~ 200–300, ~ 450–550, ~ 950–1080, ~ 1300, ~ 1600 AD. From ~ 1800 AD, decreasing $\delta^{18}O$ values of both species are recorded. The $\delta^{18}O$ values of *G. uvula* are stable throughout the entire record except lower values at ~ 340, ~ 920, ~ 1270, ~ 1400, ~ 1800 and ~ 1950 AD (Fig. 7Ic).

The $\delta^{13}C$ values of *N. pachyderma*, *T. quinqueloba* and *G. uvula* indicate generally

concurrent trends until ~ 1800 AD (Fig. 7Id–f). Between 1800 AD and the present, the $\delta^{13}C$ values of *N. pachyderma* and *T. quinqueloba* display high magnitude changes. High $\delta^{13}C$ values of *N. pachyderma* are recorded at ~ 700, ~ 1380–1440, ~ 1750, ~ 1820, ~ 1960 AD and high $\delta^{13}C$ values of *T. quinqueloba* are recorded at ~ 450, ~ 800, ~ 1300–1380, ~ 1800 and ~ 1960 AD. The $\delta^{13}C$ values of *G. uvula* are rather stable between ~ 21 and 400 AD. From ~

400 AD to ~ 1760 AD, the $\delta^{13}C$ values are generally low except for high values around ~ 1340 AD. High $\delta^{13}C$ values of *G. uvula* describe the period between ~ 1800 AD and the present.

The $\delta^{18}O$ and $\delta^{13}C$ values of *N. pachyderma*, *T. quinqueloba* and *G. uvula* reveal three different clusters (Fig. 7II). *Globigerinita uvula* shows lowest $\delta^{18}O$ values (0.30–1.48 ‰) and

lowest $\delta^{13}C$ values (–2.45 and –0.96 ‰), while *N. pachyderma* indicates highest $\delta^{18}O$ values (2.77–3.35 ‰) and highest $\delta^{13}C$ values (–0.03 and 0.23 ‰). The values of *T. quinqueloba* $\delta^{18}O$ and $\delta^{13}C$ are intermediate ranging 2.28–2.69 ‰ and –1.35 to –0.64 ‰, respectively. The offset between average values of $\delta^{18}O_{Np}$ - $\delta^{18}O_{Tq}$ is 0.6 ‰, and between $\delta^{18}O_{Tq}$ - $\delta^{18}O_{Gu}$, 1.39 ‰. Offset between average values of $\delta^{13}C_{Np}$ and $\delta^{13}C_{Tq}$ is –2.48 ‰, and between $\delta^{13}C_{Tq}$ and

$\delta^{13}C_{Gu}$, 0.68 ‰.

### 3.2.4 SST





Sea surface temperatures based on Mg/Ca ratios (SST$_{Mg/Ca}$) and $\delta^{18}$O (both *T. quinqueloba* and *N. pachyderma*) vary between 3.1 and 6.2 °C, and 3.4 and 6.6 °C, respectively (Fig. 8a,b). These temperatures are very similar to temperatures (3.9–5.4 °C) recorded at 100–250 m water depth by the three CTD casts taken during key-seasons for

reproduction of planktic foraminifera at the core site (Figs. 2Ia,b,8a,c).

Comparison of the SST reconstructed on $\delta^{18}$O$_{Np}$ and $\delta^{18}$O$_{Tq}$ show an average offset of 1.3 °C (0.3–2.6 °C). SST based on $\delta^{18}$O *T. quinqueloba* clearly show lower temperatures throughout the entire record. The SST-$\delta^{18}$O$_{Np}$ vary between 4.7 and 6.6 °C and the SST-$\delta^{18}$O$_{Tq}$ show more variable temperatures oscillating between 3.4 and 5.5 °C (Fig. 8a,b). The

sea surface temperatures obtained by transfer functions (SST$_{TF}$) ranges between 2.0 and 7.0 °C (Fig. 8c). Highest temperatures in the record are noted for the last century. As presented above (sect. 3.1.3.), the modern main depth habitats of the studied three species, we refer to surface 0– ~ 30 m as the living depth range for *G. uvula*, near surface ~ 30–100 m as the living depth range for *T. quinqueloba*, and subsurface ~ 100–250 m as the living depth range

for *N. pachyderma*. Further, we assume that the SST estimates (SST$_{Mg/Ca}$ and SST-$\delta^{18}$O$_{Np}$) and other proxies based on *N. pachyderma* refer to subsurface water masses, the SST estimates (SST-$\delta^{18}$O$_{Tq}$) based on *T. quinqueloba* refer to temperatures of the near surface water masses, and the SST$_{TF}$ estimated for 10 m water depth is associated with the surface water masses.

**4 Discussion**

**4.1 Validation and suitability of the stable isotope SST-reconstructions on *N. pachyderma*, *T. quinqueloba* and *G. uvula***

As foraminifera precipitate their calcite from the surrounding seawater it is expected that the $\delta^{18}$O value records the isotopic equilibrium of the seawater and the temperature at the calcification depth of a particular species (Emiliani, 1954). However, due to the differential isotopic uptake in calcareous organisms compared to equilibrium conditions, an offset between the equilibrium calcite value and the measured value of foraminiferal shells is found

(Erez, 1978). This offset (the "vital effect") is species-specific and varies strongly regionally (Ezard et al., 2015 and references herein). Given the discrepancy and different magnitudes of vital effects reported in the Fram Strait on $\delta^{18}$O and $\delta^{13}$C of *N. pachyderma* and *T. quinqueloba* (Table 2), and to our knowledge yet non-existing isotope data performed on *G.*





*uvula*, we do not apply any correction of vital effects. Hence, we rather interpret our oxygen and carbon isotope records cautiously addressing general trends rather than absolute numbers. In addition, we validate the suitability of the $\delta^{18}O$ and $\delta^{13}C$ records by briefly discussing other potential main effects influencing the carbon and oxygen isotopes signatures recorded in

shells of the three species studied here.

Planktic foraminiferal interspecific isotopic relationships reflect the isotopic structure of the water column. Shallow dwelling species typically have relatively low $\delta^{18}O$ values, compared to deeper dwelling species (Ravelo and Fairbanks, 1992; 1995; Coxall et al., 2007; Birch et al., 2013). In our record, the increase of $\delta^{18}O$ values from the surface species *G.*

*uvula,* through near surface dwelling *T. quinqueloba* to subsurface living *N. pachyderma* is well documented and can simply be ascribed to the salinity increase with depth (lowest values at the surface) (Fig. 7Ia–c). In contrast to the trend in $\delta^{18}O$ values, $\delta^{13}C$ values are expected to increase from subsurface to surface dwelling species, because high primary production and photosynthesis remove the $^{12}C$ and enriches the surface water masses in $^{13}C$ (Fogel and

Cifuentes, 1993). Correspondingly, the $\delta^{13}C$ of *G. uvula*, as a shallow dweller, should be described by higher values (enriched in $^{13}C$) compared to *T. quinqueloba* and *N. pachyderma*. However, our records show an opposite trend to what would be expected (Fig. 7Id–f). As the $^{13}C$ is a function of dissolved inorganic carbon in the water (Duplessy, 1978) that can be influenced by array of physical, chemical and biological factors (e.g., Spero et al., 1997;

Bemis et al., 2000; Peeters et al., 2002; Chierchi and Franson, 2009), it is very difficult to explain these inverted values in the dynamic conditions of the Fram Strait. In case of *T. quinqueloba* and *N. pachyderma* the variable $\delta^{13}C$ values could also be well associated with the species- and depth- specific isotopic uptake compared to equilibrium conditions (Table 2).

**4.2 Sea surface reconstruction, regional comparison, and climatic inferences**

There is no consensus on the precise dating, spatial extent, local occurrence and the magnitude of climatic changes that occurred in the last two millennia in the Northern Hemisphere (IPCC, 2007). Therefore, we discuss our data on the basis of major changes in

our proxy records (concentration of foraminifera, species distributions, shell weights, isotope records of *N. pachyderma* and *T. quinqueloba*, and SST (transfer function and $\delta^{18}O_{Tq}$), which correlate closely in time with climatic and historical periods defined in Europe by Lamb (1977). Moreover, we interpret small/large differences in $\Delta\delta^{18}O_{Np-Tq}$ as an indicator of relative





increased/decreased influence of warm and salty Atlantic Waters in the subsurface to near surface water masses and the $\Delta\delta^{18}O_{Tq-Gu}$ as an indicator of increased/decreased freshening - relative presence of melt/polar water and a measure of reinforcement of stratification in the surface. In addition, we apply the percentage abundance of *G. uvula* as an independent

indicator of fresh, low salinity surface waters (Hemleben et al., 1989; Boltovskoy and Lena, 1970b) (Fig. 8d–f).

**4.2.1 ~ 21 – ~ 400 AD, the second half of the Roman Warm Period, RWP**

High relative abundance of the subpolar species *T. quinqeloba*, which is indicative of warm and saline Atlantic Water inflow (Volkmann, 2000), together with high concentration of planktic foraminifera suggest favourable conditions at the sea surface (Fig. 4f,h). The mean shell weights of both juvenile (100–125 µm) and adults (150–180 µm) forms are high and the % fragmentation is low indicting well-saturated with respect to calcium carbonate Atlantic

Water masses (e.g., Huber et al., 2000), except for a short-lasting interval at ~ 140 AD, when the % fragmentation increased (Fig. 6). The mean shell weight of *N. pachyderma* (150–180 µg) reaches its maximum of 3.32 µg around ~ 300 AD which further indicate strong influence of Atlantic Water during this period. Generally, high mean shell weights of all species, low %TOC, and other parameters indicate a good preservation for all species, similar to ~ 317 km

north of our core site at ~ 0–200 AD (Zamelczyk et al., 2013). Preservation patterns at these both sites point to a strong influence of calcium carbonate-rich Atlantic water masses in the eastern Fram Strait (e.g., Huber et al., 2000) during the second half of the Roman Warm Period.

All SST estimates unanimously indicate presence of warm and saline water masses

(Fig. 8a–c). The $\delta^{18}O$ values of *N. pachyderma* and *T. quinqeloba* are low but relatively high in *G. uvula* possibly indicating thermal or saline decoupling of the surface water masses. At ~ 200–350 AD, the SST are highest, suggesting the influence of Atlantic water masses to be strongest (Fig. 8a–c). Also, the elevated $\Delta\delta^{18}O_{Np-Tq}$ indicate increased inflow of Atlantic Water in the subsurface and near surface (Fig. 8d). During the same time interval, the

$\Delta\delta^{18}O_{Tq-Gu}$ and the %*G. uvula* are high pointing to episodically increased stratification (Fig. 8e,f). Moreover, the $\delta^{13}C_{Gu}$ values indicate constantly productive water masses in the upper ocean, whereas the $\delta^{13}C_{Np}$ and $\delta^{13}C_{Tq}$ records point to highly variable conditions in near



surface and subsurface water masses during this time (Fig. 7d–f). This may indicate that the stratification could have been associated with the proximity of the sea-ice margin.

The warm surface conditions during this time are in agreement with other studies from the North Atlantic Ocean. South of Iceland, in the eastern Labrador Sea and where the North Atlantic Current enters the Fram Strait, enhanced inflow of warm and salty Atlantic Waters is inferred from high differences between the $\delta^{18}$O measured on *N. pachyderma* and *T. quinqueloba*, low % of *N. pachyderma* and reduced sea-ice cover (Moffa-Sánchez and Hall, 2017). In the North, on the continental margin off western Svalbard, high planktic foraminiferal fluxes and increased subsurface and bottom water temperatures are linked to an increase in Atlantic Water heat and volume transport (Werner et al., 2011).

### 4.2.2 ~ 400 – ~ 800 AD, the Dark Ages Cold Period, DACP

During this time interval, the concentration of planktic foraminifera decreased markedly and the percentages of subpolar *T. quinqueloba* is low (Fig. 5f,g–j). Together with the dominance of the cold-water species *N. pachyderma* this indicate change towards cold oceanic conditions (Fig. 4g–j). The mean shell weights of *N. pachyderma* and *T. quinqueloba* decrease, which indicate a deterioration of the calcification conditions for these species (Fig. 6a). At ~ 610 AD, the $SST_{Mg/ca}$ and the $SST_{TF}$ show a ~ 1 ºC drop in temperature of subsurface and surface water masses, compared to the previous period (Fig. 8a,c). The low $\Delta\delta^{18}O_{Np-Tq}$ suggest reduced Atlantic Water influence in the subsurface to near surface (Fig. 8d). This coincides in timing with cool surface conditions indicated by planktic foraminiferal fluxes and dinocyst assemblages north of our study area (Werner et al., 2011; Bonnet et al., 2010).

At ~ 700 AD, the decreased $\Delta\delta^{18}O_{Np-Tq}$ and increased $\Delta\delta^{18}O_{Tq-Gu}$ indicate clearly a weak Atlantic water inflow and freshening at the surface that possibly prevented vertical mixing (Fig. 8d,f). Although, the less saline conditions in the near surface and surface are indicated by single data points, they are in agreement with other, low resolution, studies that report stratification of surface water masses and cooler upper surface-water for the late Holocene in this region (Berben et al. 2014; Werner et al., 2016). Increased sea-ice cover and reduced influence of Atlantic inflow with a southern advancement of the polar front has likely been responsible for the freshening of surface waters during this time (Berben et al. 2014; Werner et al., 2016; Moffa-Sánchez and Hall, 2017).





The SST$_{\delta^{18}ONp}$ and SST$_{Mg/Ca}$ deviate from each other in this interval. While SST$_{\delta^{18}ONp}$ and SST$_{\delta^{18}OTq}$ are somewhat similar SST$_{Mg/Ca}$ is opposite in trend (Fig. 8a,c). This discrepancy might be linked to the salinity effect on Mg/Ca of the water masses in which *N. pachyderma* migrates during its ontogeny (Spielhagen and Erlenkeuser, 1994; Schiebel and Hemleben, 2005).

The $\delta^{18}O_{Np}$ and $\delta^{18}O_{Tq}$ values are low at ~ 400–600 AD and increase towards the present (Fig. 7Ia,b). In contrast, the $\delta^{18}O_{Gu}$ show relatively constant values suggesting no change in isotopic composition of the surface water masses possibly indicating a thermal or saline decoupling of the surface layer from the near surface and subsurface water masses (Fig. 7Ic).

Overall, taking into account differences in sedimentation rates, dating control, and marine reservoir age corrections, a warming during the RWP and the subsequent cooling during DACP in the Storfjorden Fan can be considered as a widespread phenomenon occurring in the Norwegian Sea and in the eastern North Atlantic with similar timing (e.g. Bond et al., 2001; Andersson et al., 2003).

### 4.2.3 ~ 800 – ~ 1400 AD, the Medieval Warm Period, MWP

The concentration of planktic foraminifera during this interval reaches its maximum of the 2000 year long record (Fig. 4f). The cold species *N. pachyderma* decreases and the subpolar species show high percentages indicting change to characteristic foraminiferal assemblage for warm conditions (Fig. 4g–j). At ~ 900–1400 AD, the average ocean SST$_{TF}$ is ~ 5.9 °C (Fig. 8c). The near surface and subsurface water masses experience a temperature rise as indicated by SST$_{Mg/Ca}$ and SST$_{\delta^{18}O}$ on *N. pachyderma* and SST$_{\delta^{18}O}$ *T. quinqueloba* between ~ 900 and ~ 1100 AD (Fig. 8a,b). Also, the high $\Delta\delta^{18}O_{Np-Tq}$ points to strong Atlantic Water inflow in the same time (Fig. 8d). All these proxies clearly suggest a dominance of Atlantic water masses and discernible warm and favourable sea surface conditions for planktic foraminifera at our coring site.

The warming is also expressed by the gradual increase in mean shell weight of both, the juvenile and the adult forms of *N. pachyderma* and *T. quinqueloba* and low shell fragmentation (Fig. 6a,b). Consistently, in the north-eastern Fram Strait at 78 °N, high mean shell weights of *N. pachyderma,* low fragmentation and other dissolution proxies suggests favourable sea surface conditions at this time (Zamelczyk et al., 2013).





During the MWP, the trends of $\delta^{13}$C in *N. pachyderma*, *T. quinqueloba* and *G. uvula* are in approximate accordance (Fig. 7d–f). This might indicate that the water masses in the surface, near surface and subsurface became similarly ventilated and productive. However, the divergent trends of $\delta^{18}$O values in the three species suggest that the water column remained thermally or salinity stratified most of the time, even during the strongest inflow of Atlantic Water at ~ 900–1100 AD as indicated by high $\Delta\delta^{18}O_{Np-Tq}$, $SST_{TF}$ and $SST_{\delta^{18}OTq}$ (Figs. 7a–c,8b–d). At the same time, low % of *G. uvula* and $\Delta\delta^{18}O_{Tq-Gu}$ points to well-mixed waters with the absence of a low salinity water layer at the surface indicating ice-free conditions during most of the year. Similar warm conditions reported on the West Spitsbergen slope and around Iceland were linked to the strengthened inflow of Atlantic Water that was probably accountable for ice-free winters and consequently a lack of stratification during spring in the Atlantic Water zone in the Fram Strait (Werner et al., 2011; Moffa-Sánchez and Hall, 2017).

The period between ~ 1300 and ~ 1500 AD is characterized by high-magnitude changes in most of our proxies suggesting highly variable subsurface, near surface and surface water conditions. Sudden shifts in all reconstructed SSTs, $\delta^{18}$O and $\delta^{13}$C values, as well as the short lasting stratification as indicated by $\Delta\delta^{18}O_{Tq-Gu}$ and increased abundance of *G. uvula* point to an enhanced influence of fresh melt water at the surface (Figs. 7I,8e,f). This was probably caused by increase in sea-ice cover during winters initiated by the transition to the cooler sea surface conditions. A similar, interrupted by abrupt decadal-scale warmings, transition from the MWP to the LIA is commonly recognized in the proxy records in the Arctic and Subarctic region (Werner et al., 2017).

Numerous studies have documented warmer sea surface conditions as a local expression of a widely occurring warming since ~ 900 AD in the North Atlantic (e.g., Dahl-Jensen et al., 1998; Andersson et al., 2003; Moberg et al., 2005; Bonnet et al., 2010; Werner et al., 2011; Zamelczyk et al., 2013; Miettinen et al., 2015). The Arctic surface air temperature (SAT) reconstruction and the Northern Hemisphere spatial SAT identify the MWP as ~ 300 year long interval from 950 to ~ 1250 AD (Mann et al., 2009; Kaufman et al., 2009). The Greenland Summit station $\delta^{18}$O data at 950 AD suggest warmer temperatures in Greenland during this time (Vinther et al., 2010). Overall, the patterns of reconstructed proxies in core HH12-1206BC agree very well with the spatial SAT reconstruction by Mann et al. (2009) during the MWP. The temperature drop indicated by $SST_{TF,}$ shifts in $\delta^{18}$O and $\delta^{13}$C values, mean shell weights, and $\Delta\delta^{18}O_{Np-Tq}$ and $\Delta\delta^{18}O_{Tq-Gu}$ around ~ 1300–1450 AD are





also shown as a distinct temperature anomaly in the SAT reconstructions of Mann's (2009)
for the entire Northern Hemisphere at the same time.

### 4.2.4 ~ 1400 – ~ 1950 AD, the Little Ice Age, LIA

The decrease of foraminiferal concentration to a minimum in the entire record in
addition to the dominance of cold species *N. pachyderma* signalize the transition to cold
conditions of the LIA (Fig. 4f,g). During the period ~ 1400–1700 AD, the mean shell weight
first decreased slightly and at ~ 1550–1700 AD reached high values indicating variable

calcification capability (Fig. 6a). At ~ 1520 AD, the % shell fragmentation was high
indicating a tendency to shell breakdown (Fig. 6b). We cannot rule out that this short-lasting
change represents deteriorating conditions for preservation. However, this seems to be
unlikely, as the % TOC and $CaCO_3$ remained stable. Therefore we link the reduced intensity
of calcification and fragility of the planktic foraminiferal shells to changes in $[CO_3^{2-}]$ and $\Omega_{Ca}$

due to sea-ice melting during peak reproduction months in summers (Manno et al., 2012;
Fransson et al., 2013). This is supported by the ~ 1.3 ºC temperature drop in surface water
masses indicated by $SST_{TF}$ (Fig. 8c). Whereas the surface experienced a slight cooling, the
near surface and subsurface water masses warmed markedly as shown by the increased
$SST_{\delta^{18}OTq}$ and $SST_{Mg/Ca}$, respectively (Fig. 8a,b). The Atlantic Water inflow was strong, as

indicated by the high $\Delta\delta^{18}O_{Np\text{-}Tq}$, providing calcium carbonate-rich, well-saturated water
masses (Fig. 8d; Huber et al., 2000) and possibly amplifying the sea-ice melting process.

The time interval ~ 1600–1800 AD, was characterized by slightly higher concentration
of planktic foraminifera, lower percentages of *N. pachyderma*, relatively high relative
abundances of subpolar species (*T. quinqueloba*, *G. uvula*, *N. incompta*) and high shell

weights indicting warming of the surface ocean (Figs. 4f–j,6a). The $SST_{TF}$ were at the same
level as the $SST_{TF}$ during the MWP pointing to warm surface conditions (Fig. 8c). However,
the inflow of Atlantic water diminished as indicated by the high $\Delta\delta^{18}O_{Np\text{-}Tq}$ and the surface
and subsurface water masses experienced a slight cooling indicted by $SST_{\delta^{18}OTq}$ and $SST_{Mg/Ca}$,
respectively (Fig. 8a,b,d). In the end of the warm phase, at ~ 1790 AD, the $\Delta\delta^{18}O_{Tq\text{-}Gu}$ indicate

increased stratification (Fig. 8f).

This warmer period within the LIA interval has also been reported in studies from the
North Atlantic occurring at 1600–1820 AD (Miettinen et al., 2015) and at 1700–1800 AD
(e.g., Mann et al., 2009). However, the warm phase at 1600–1800 AD at Storfjorden Fan does





not initiate the progressive shift to the warm conditions of the last century, as the following
interval at ~ 1800–1950 AD was characterized by the lowest SST of the past two millennia
(Fig. 8a–c).

Between ~ 1800 and ~ 1950 AD, a shift to very low foraminiferal concentrations with
dominance of *N. pachyderma* (up to 88 %), very low mean shell weights and high %
fragmentation and markedly increased IRD concentration indicate the change to harsh ocean
conditions with enhanced sea-ice and iceberg rafting (Figs. 4a,b,f,g,6a,b). The lowest SST
($SST_{TF}$ ~ 2 °C, $SST_{Mg/Ca}$ ~ 3.3 °C, $SST_{\delta^{18}O_{Tq}}$, ~ 3.1 °C) and percentages of subpolar species
point to minimum ocean temperatures during the past 2000 years (Figs. 4h–j,8a–c). Also, the
accordance of high values and trends of $\delta^{18}O$ and $\delta^{13}C$ in *N. pachyderma*, *T. quinqueloba* and
*G. uvula,* may suggest a common depth habitat indicative of harsh sea surface conditions as
under permanent sea-ice cover, *N. pachyderma* and *T. quinqueloba* may migrate to better
ventilated, nutrient-rich surface waters (Volkmann, 2000) (Fig. 7I).

**4.2.5 ~ 1950 AD, the last century**

Since ~ 1950 AD, the foraminiferal concentration and the dominance of *T.*
*quinqueloba* and *G. uvula* increased gradually towards the present indicting progressive
warming (Fig. 4h–j). At the core top, the % of the cold species *N. pachyderama* decreases to
its minimum of the entire record (Fig. 4g). Although the $\Delta\delta^{18}O_{Np-Tq}$ indicate low Atlantic
Water inflow, the sea subsurface conditions during the last century clearly experienced
warming, as indicated by $SST_{TF}$ (Fig. 8c,d). The highest abundance of *G. uvula* and high IRD
deposition suggest freshening of the near surface of the ocean, but probably not sufficient to
develop a stratification as indicated by low $\Delta\delta^{18}O_{Tq-Gu}$ (Figs. 4a,i,8e,f). Whereas the upper
surface ocean of the past ~ 60 years reaches a maximum of ~ 7 °C, with an average of 5.4 °C
as indicated by $SST_{TF}$, the temperatures of the near surface and subsurface water masses seem
to increase only slightly, which is suggested by the $SST_{Mg/Ca}$ and $SST_{\delta^{18}O}$ measured on the
subsurface dwelling *N. pachyderma* and near surface dwelling *T. quinqueloba* (Fig. 8a–c).

As the concentration of planktic foraminifera increased and the surface ocean shifted
towards warm and more favorable growth conditions, an increase in shell weights of *N.*
*pachyderma* and *T. quinqueloba* would be expected. However, the mean shell weights show
only a slight weight increase and no change in % fragmentation (Fig. 6a,b). It has been shown
that temperature and the carbonate saturation play an important role in the calcification of





foraminiferal shells (Russell et al., 2004; Lombard et al., 2010; Manno et al., 2012). Under low concentration of $[CO_3^{2-}]$, shell calcification rates tend to decrease (Manno et al., 2012). This can be linked to the ocean acidification where increased $CO_2$ in the ocean leads to a decrease of $[CO_3^{2-}]$ and calcium-carbonate saturation (e.g. Chierici et al., 2011)

There is a paleoproxy consensus of the progressive warming over the last ~ 200 years in the Arctic region (e.g., Kaufman et al., 2009; Majewski et al., 2009; Spielhagen et al., 2011; Werner et al., 2011; Zamelczyk et al., 2013). Some studies consider the past 100 years as the warmest period of the past two millennia (Spielhagen et al., 2011; Werner et al., 2011), although preservation issues with improved preservation in surface sediments may have

biased results (Zamelczyk et al., 2013). At Storfjorden Fan, the $SST_{TF}$ reconstructions show a distinct warming trend at the surface, while the $SST_{Mg/Ca}$ and the $\delta^{18}O$ on *N. pachyderma* and *T. quinqueloba* do not show any near or subsurface warming. This inconsistency might be linked to a different source and/or mechanism for the thermal change in the water masses or preservation.

### 4.3 Solar irradiance - driver of short-scale changes in sea surface conditions in the late Holocene?

        The environmental controls of the foraminiferal-based proxies at the vicinity of
Storfjorden Fan and the west Spitzbergen shelf during the late Holocene are clearly a function of the strength of the Atlantic Water inflow to the Fram Strait and the proximity of the sea-ice margin (e.g., Risebrobakken et al., 2003; 2011; Hald et al., 2007; Rasmussen et al., 2007, 2013; Majewski et al., 2009; Zamelczyk et al., 2013; Aagaard-Sørensen et al., 2014; Berben et al., 2014; Werner et al., 2013, 2016; Łącka et al., 2016; Telesiński et al., 2017; this study).

However, several previous studies suggested significant influence of solar irradiance on North Atlantic climate throughout the Holocene (Lamb, 1979; Bond et al. 2001).
        We compare our foraminiferal near sea surface temperatures at 10 m water depth based on transfer functions and the relative abundances of the surface dwelling *G. uvula* with the total solar irradiance (TSI) reconstructed from $^{10}Be$ in Antarctic and Greenland ice cores

(Steinhilber et al., 2012). Despite of the age model differences of the records, the timing of SST shifts in Storfjorden Fan show a marked correlation with total solar irradiance variability superimposed on the large-scale, multi centennial warm and cool periods, that correlate with the well-known climatic events of the last two millennia (RWP, DACP, MWP, LIA) (Fig. 9).




Most of the phases of solar minima correspond to low SST and low % of *G. uvula* and vice versa, solar maxima match with high SST and high abundances of *G. uvula* (Fig. 9a–c).

The Subpolar Gyre is considered a dominant large-scale feature of the surface circulation of the North Atlantic (Hatun et al., 2005; Higginson et al., 2011). In the Iceland

Basin, Moffa-Sánchez et al. (2014) reconstructed SST and salinity using paired $\delta^{18}$O and Mg/Ca ratio on near surface dwelling *Globorotalia inflata,* (Rebotim et al., 2016) and climate model simulation. The authors suggest a strong correlation between SST and TSI over the past 1000 years has been linked to the strength of the Subpolar Gyre and Atlantic persistent atmospheric blocking events. Jiang et al. (2015) shows equally strong correlation between

SST reconstructed on diatoms assemblages and TSI which supports further the SST-TSI linkage in this region. Numerous modeling studies postulate a strong influence of the Subpolar Gyre also on the Atlantic Water inflow to the Fram Strait and the Barents Sea (Karcher et al. 2003; Hatun et al., 2005; Lohmann et al., 2009).

The correspondence of sea surface temperatures recorded by planktic foraminifera at

Storfjorden Fan during known TSI anomalies of the last 1000 years imply that solar activity could have had a dominant influence on sea surface conditions on the decadal to multidecadal scale (Fig. 9b,c). Moreover, the link between the high relative abundance of *G. uvula*, a low salinity water species, and the high TSI suggest an influence of solar irradiance on the intensity of stratification formed most likely by solar heating of the upper ocean layers (Fig.

9a,c). However, the different resolutions of the two records hamper the precise comparison and make it impossible to determine the forcing of sea surface condition changes at the HH12-1206BC site. Despite this hindrance, the results might suggest that at least over the last millennium solar forcing, possibly amplified by atmospheric forcing, has been responsible for the short-term variability of the surface conditions superimposed on the multi-centennial

warm and cool periods at Storfjorden Fan.

Despite the increasing numbers of paleorecords proposing the TSI-Holocene SST connection in high-latitude regions of the North Atlantic (Bond et al., 2001; Jiang et al., 2015; Moffa-Sanchez et al., 2014) as well as of the North Pacific (Clegg et al., 2011; Tinner et al., 2015), the linkage of late Holocene climate change to solar irradiance is controversial as

several studies have attributed volcanism a considerably more important factor than solar irradiance e.g. during the LIA (Hegerl et al., 2003; Miller et al., 2012). Other studies considered both volcanic and solar forcing responsible for the global mean LIA signal, but emphasized the volcanic forcing as a major player in global-scale cooling, and the solar





variability as a major player in the regional changes and extreme cold events (Shindell et al., 2003).

Until recently, sea surface temperatures and climate variability on multidecadal timescales were largely believed to be controlled by internal processes rather than external
forcing. However, the foraminiferal SST-TSI correlation suggest that the solar variability is affecting short-term sea surface conditions through ocean-atmosphere dynamics not only in the area of the Supolar Gyre, but also further north in the northern Fram Strait. This potential linkage warrants further research in order to understand this component of natural climate variability, which apparently exerts an essential influence on the northern North Atlantic
climate system.

**5 Conclusions**

The major climate anomalies of the past two millennia were reflected in planktic foraminiferal records of the Storfjorden Fan (distribution of species, SST reconstructions by
Mg/Ca, transfer functions and trend of $\delta^{18}$O and $\delta^{13}$C values of *N. pachyderma*, *T. quinqueloba* and *G. uvula*). The oceanic conditions at ~ 400–800 AD and ~ 1400–1950 AD are associated with the local expression of the Dark Age Cold Period (DACP) and the Little Ice Age (LIA), respectively. They are characterized by low concentrations of planktic foraminifera, low relative abundances of subpolar species and dominance of the polar species
*N. pachyderma*. The surface (30–0 m) conditions were cold, but although the Atlantic water influence diminished, the near surface (100–30 m) and subsurface (250–100 m) were described by relatively high temperatures. During the LIA, the period of ~ 1600–1800 AD, was relatively warm but it was followed by the coldest interval of the past two millennia at ~ 1800–1950 AD.

High concentration of planktic foraminifera, high percentages of subpolar species and decreased abundances of *N. pachyderma*, with high mean shell weight and high temperatures identified periods at ~ 0–400 AD, ~ 800–1400 AD and from ~ 1950 AD to the present which have been linked to the RWP, MWP and the most recent warming, respectively.
The reconstructed properties and structure of the water column seem to be a function of the
inflow of Atlantic water masses and presence of see ice as well as the variability of the location of sea ice margin. Strongest Atlantic Water inflow and highest temperatures in the upper water column (subsurface, near surface and surface) water masses occurred during MWP at ~ 900–1100 AD.



Stratification of the surface water masses, inferred form the trends in $\delta^{18}O$ values measured on the surface-dwelling *G. uvula*, near surface-dwelling *T. quinqueloba* and subsurface-living *N. pachyderma* likely persisted over the past 2000 years. This is in agreement with other studies, which postulate stratification reinforcement during the Late

Holocene.

The short lasting minima of SST calculated by transfer functions in the surface waters and decreased relative abundance of *G. uvula*, the low-salinity surface water species, are in phase with solar minima. These short-lasting changes are superimposed on the large-scale, multi-centennial warm and cool periods that correlate with the above-mentioned, well-known

climatic events of the last two millennia and imply that solar forcing has been responsible for decadal to multidecadal sea surface variability over the Storfjorden Fan over the last two millennia.

*Data availability.* The foraminifera data set will be made publicly available on PANGAEA

online data archives.

*Author contributions.* KZ performed the core processing and foraminifera analyses and initiated the study in correspondence with TLR. MR provided the trace elements data and MC carried out water carbonate chemistry analyses. All authors discussed and edited the

manuscript.

*Competing interests.* The authors declare that they have no conflict of interest.

**Acknowledgements**

We thank the captains and crews of *RV Jan Mayen*/*Helmer Hanssen* and S. Iversen B.R. Olsen for technical assistance by retrieving the sediment core. We also thank T. Dahl for laboratory assistance and T. Grytå for creating the area map. We appreciate B. Sternal thoughtful suggestions and discussion on interpretation of the $^{210}Pb$ and $^{137}Cs$ dating. The

study was carried out within the framework of the project „Effects of ocean chemistry changes on planktic foraminifera in the Fram Strait: Ocean Acidification from natural to anthropogenic changes" funded by The Research Council of Norway (project number 216538), the Centre for Arctic Gas Hydrate, Environment and Climate (CAGE), UiT



supported by the Research Council of Norway, project number 223259 and the Arctic
University of Norway, Tromsø.

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





**Table 1** AMS[14]C and calibrated dates in core HH12-1206BC.

| Lab nr | Depth (cm) | Age [14]C BP | Calibrated yr BP | δ[13]C | Age AD/BC |
|---|---|---|---|---|---|
| Poz-59603 | 2–3.5 | 520 ± 80* | 1–279 | −3.9 | ˜1949–1671 |
| Poz-102419 | 4.5–6 .5 | 460 ± 40* | 160–198 | −22.6 | 1752–1790 |
| Poz-57342 | 9.5–11.5 | 900 ± 80* | 357–644 | −16.2 | 1306–1593 |
| Poz-102418 | 13–14 | 580 ± 35 | 94–292 | −4.1 | 1658–1856 |
| Poz-66095 | 14–16 | 830 ± 110 | 258–634 | −14.6 | 1316–1692 |
| Poz-59605 | 19–20 | 1175 ± 30 | 652–779 | 0.3 | 1171–1298 |
| Poz-102417 | 22–22.5 | 1515 ± 30 | 966–1158 | −2.6 | 792-984 |
| Poz-66211 | 23.5–25 | 1870 ± 70 | 1272–1560 | −17.6 | 390–678 |
| Poz-57343 | 30–31 | 2320 ± 80 | 1724–2132 | −4.3 | 183BC–226 |

*Date not used in age model.

**Table 2** Compilation of vital effects reported from the Fram Strait in the literature for
*Neogloboquadrina pachyderma* and *Turborotlita quinqueloba* and references.

| N. pachyderma | | T. quinqueloba | | Reference |
|---|---|---|---|---|
| δ[18]O (‰) | δ[13]C (‰) | δ[18]O (‰) | δ[13]C (‰) | |
| −1.5 | −2.6 | −3.7 | −3.6 | Pados et al. (2015) |
| 0 | – | −0.7 | – | Jonkers et al. (2010) |
| −1 | −0.85 | −1.1 | −2 | Simstich et al. (2003) |
| −1.6 | −2.1 | −1.3 | −2.6 | Volkmann and Mensch (2001) |
| −0.9 | −1 | −1.2 | −2.2 | Stangeew (2001) |



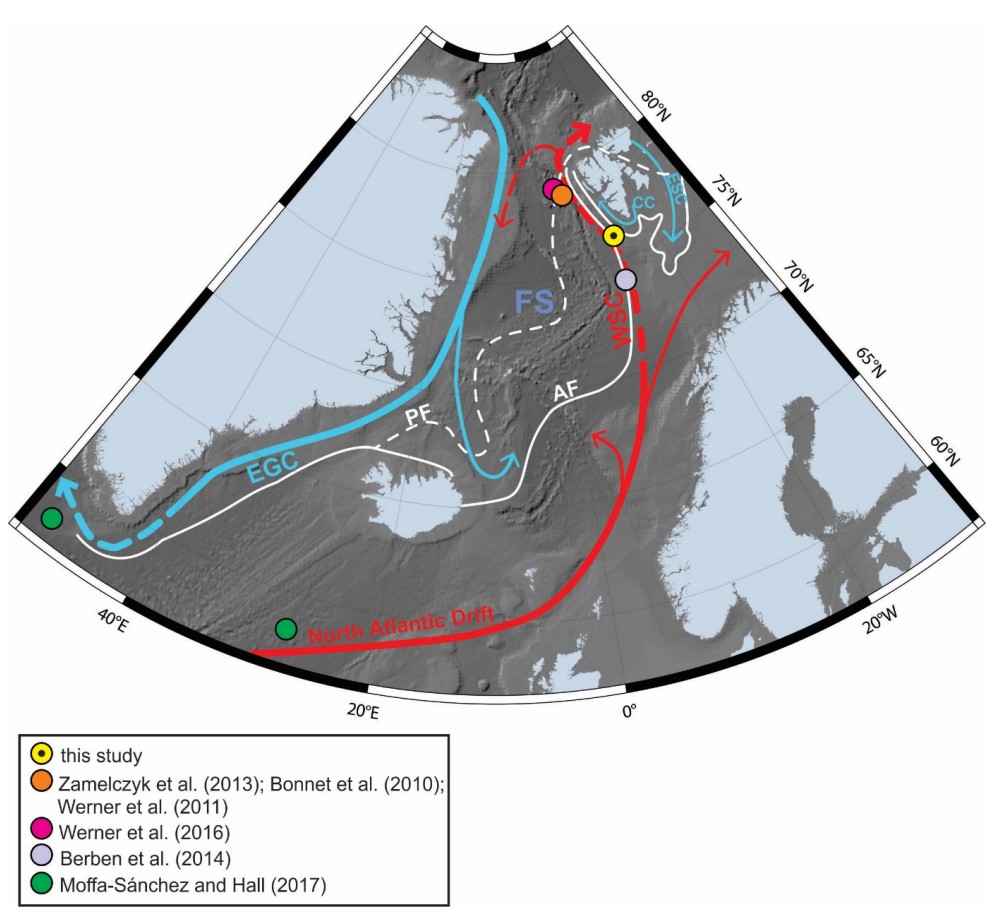

**Figure 1**

Schematic map of the northern North Atlantic and the Fram Strait showing present-day
surface currents and average position of the Polar and Arctic fronts (based on Marnela et al.
2008). The solid and dashed lines indicate surface and subsurface currents, respectively.
Location of core HH12-1206BC (yellow circle) is indicated. Abbreviations: FS: Fram Strait;
WSC: West Spitsbergen Current; CC: Coastal Current; ESC: East Spitsbergen Current; EGC:
East Greenland Current; AF: Arctic Front; PF: Polar Front. Locations of published records
discussed in the text are marked.



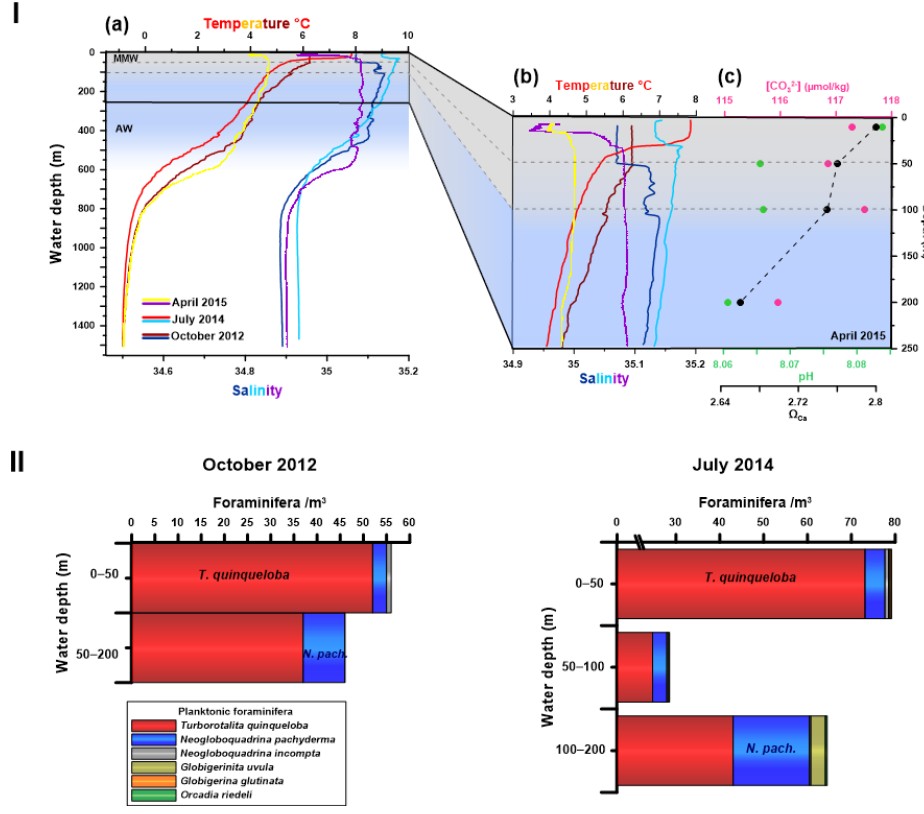

**Figure 2**

I: (a) Conductivity-temperature-depth (CTD) profile showing water mass distribution of the water column in October 2012, July 2014 and April 2015 at site HH12-1206BC; (b) close-up of CTD data in upper 250 m; (c) carbonate ion concentration $[CO_3^{2-}]$, calcium carbonate saturation state ($\Omega_{Ca}$) and pH in upper 250 m in April 2015. II: Distribution of planktic foraminifera in the water column in October 2012 and July 2014.



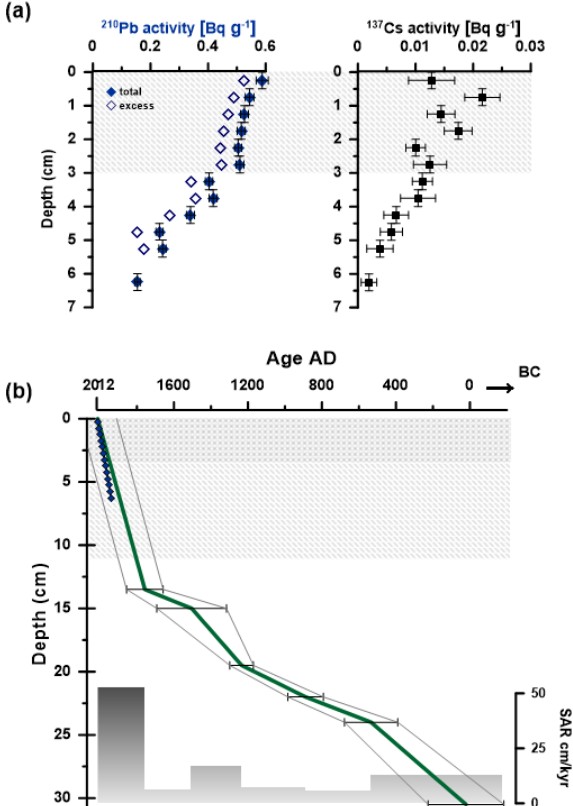

**Figure 3**

Age model based on $^{210}$Pb, $^{137}$Cs and AMS $^{14}$C datings in core HH12-1206BC: (a) total and
excess $^{210}$Pb activity profiles and $^{137}$Cs activity profile. In (a) vertical bars indicate sampling
range, whereas horizontal bars mark 2σ uncertainties. In (b) error bars of AMS dates represent
2σ standard deviation and are indicated by grey-lined field, and dark blue diamonds indicate
the age model based on $^{210}$Pb activity. Dashed area in (a) and crossed area in (b) represent
mixing of sediment in the upper 3 cm indicated by $^{210}$Pb activity profiles. In (b) dashed area
shows possible redeposition events indicated by reversed $^{14}$C dates. Sedimentation rate is
shown at the bottom of panel (b).



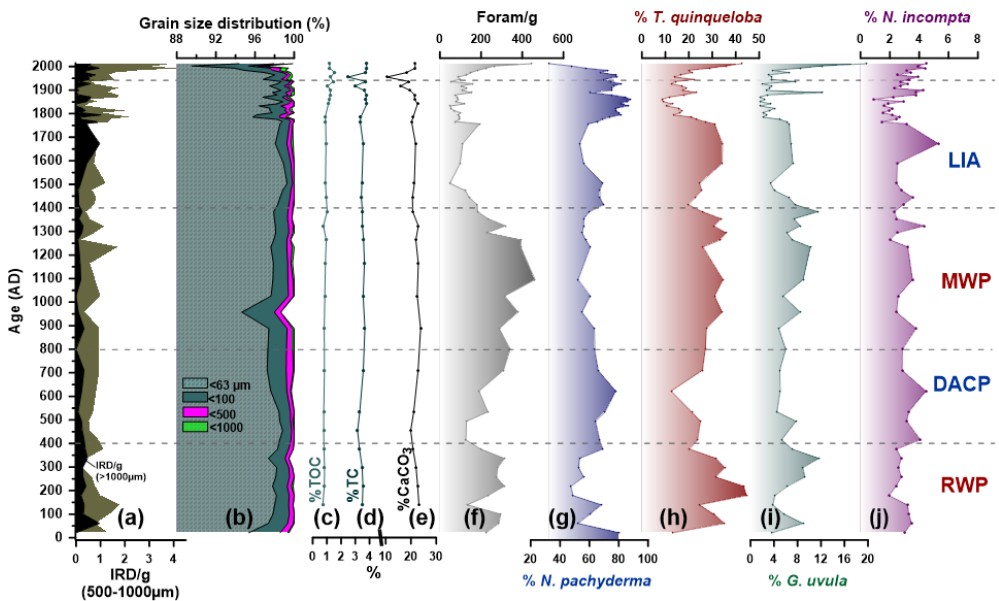

**Figure 4**

Grain size distribution, geochemical and foraminiferal records plotted versus age in core HH12-1206BC. (a) concentration of IRD in number per g dry weight sediment, (b) % cumulative diagram of grain size distribution, (c) % TOC, (d) % TC, (e) % CaCO3, (f) concentration of planktic foraminifera in number per g dry weight sediment, and relative abundance of (g) *N. pachyderma*, (h) *T. quinqueloba*, (i) *G. uvula*, (j) *N. incompta*. Abbreviations: RWP, Roman Warm Period; DACP, Dark Ages Cold Period; MWP, Medieval Warm Period; LIA, Little Ice Age.



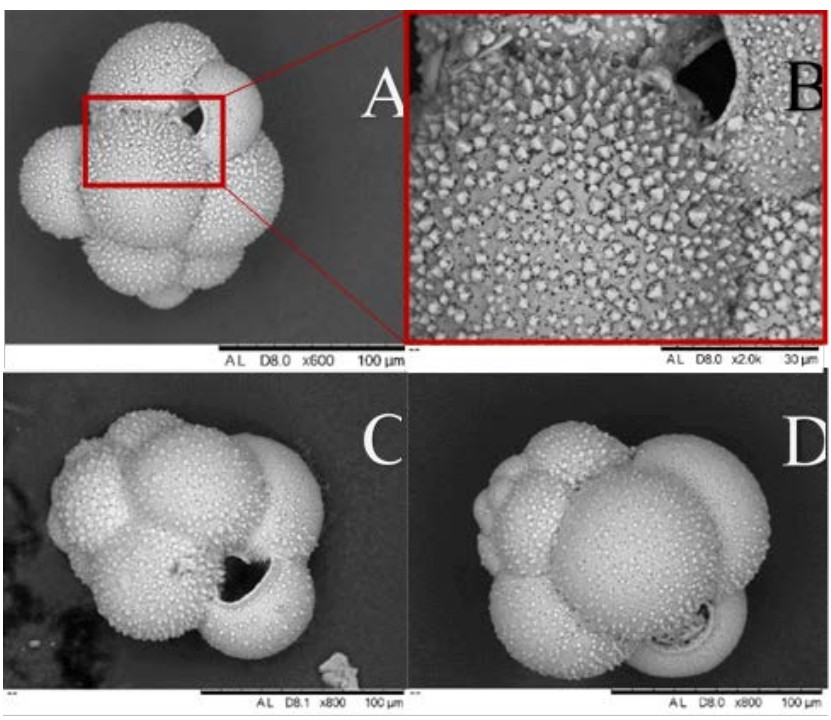

**Figure 5**

SEM pictures of planktic foraminiferal species *Globigerinita uvula minuta* (Natland, 1938) in core HH12-1206BC at A, B) 9.5–10 cm, C) 7.5–8 cm, D) 11–11.5 cm down core. This form is similar to *Globigerinita uvula,* but markedly larger, with strongly inflated chambers and comparatively low trochospiral test. Like *G. uvula* it possesses a smooth, finely perforated wall, but with stronger encrustations of crystalline knobs (B) (Rögl and Bolli, 1973).



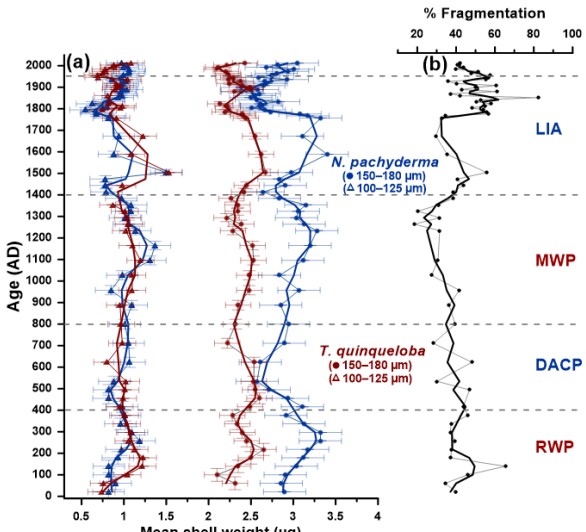

**Figure 6**

(a) Mean shell weight of planktic foraminiferal species *N. pachyderma* and *T. quinqueloba* in size-fraction 100–125 μm and 150–180 μm, (b) % fragmentation of planktic shells in core HH12-1206BC. Thick lines indicate 3-point moving average and horizontal bars mark standard deviation.



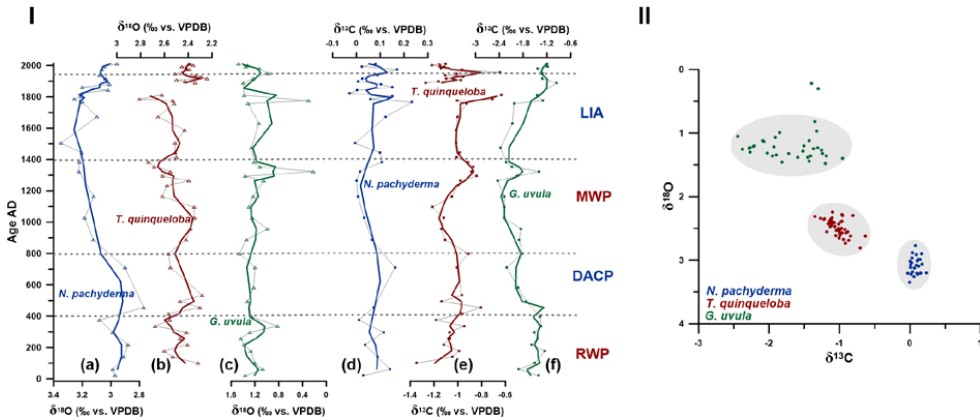

**Figure 7**

Stable isotope records (raw data) in core HH12-1206BC. I: $\delta^{18}O$ and $\delta^{13}C$ of (a,d) *N. pachyderma*, (b,e) *T. quinqueloba* and (c, f) *G. uvula*, respectively. Thick lines represent 3-point moving averages. II: Oxygen versus carbon isotope plot of *N. pachyderma*, *T. quinqueloba* and *G. uvula*.





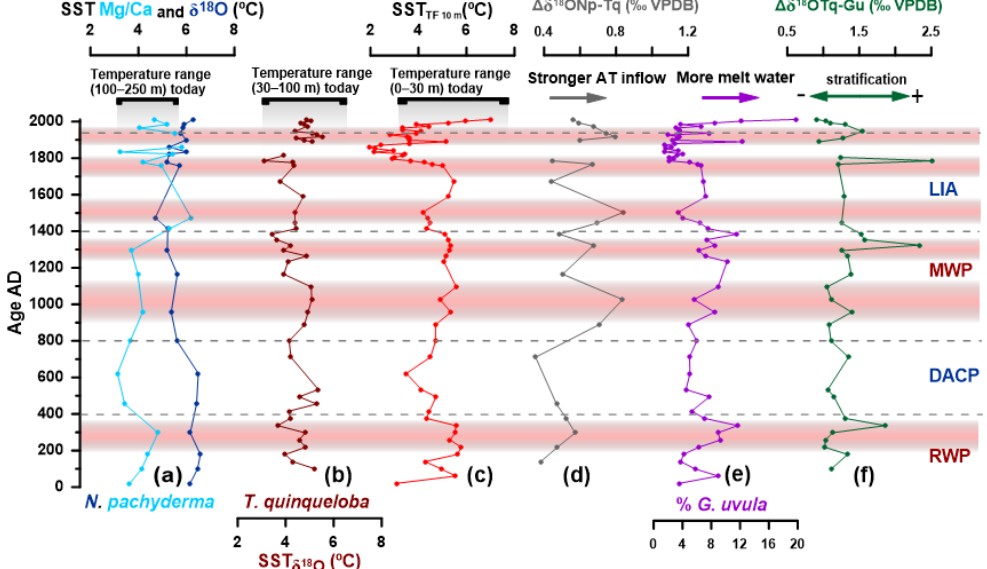

**Figure 8**

Comparison of reconstructed SST plotted versus age in core HH12-1206BC. (a) $\delta^{18}O$ and Mg/Ca ratio based SST on *N. pachyderma*, (b) $\delta^{18}O$ based SST on *T. quinqueloba*, (c) transfer function based SST$_{TF}$ at 10 m water depth, (d) $\Delta\delta^{18}O_{Np-Tq}$ as an indicator of stronger Atlantic Water inflow (e) % *G. uvula* as a proxy for supply of meltwater, and (f) $\Delta\delta^{18}O_{Tq-Gu}$ as an indicator of relative changes in stratification of near surface and surface water. Gray bars on top represent present day temperature ranges at 100–250 m, 30–100 m and 0–30 m water depth. Red bars indicate periods of stronger Atlantic Water inflow accompanied by increased presence of meltwater and enhanced stratification of the subsurface, near surface and surface water masses.




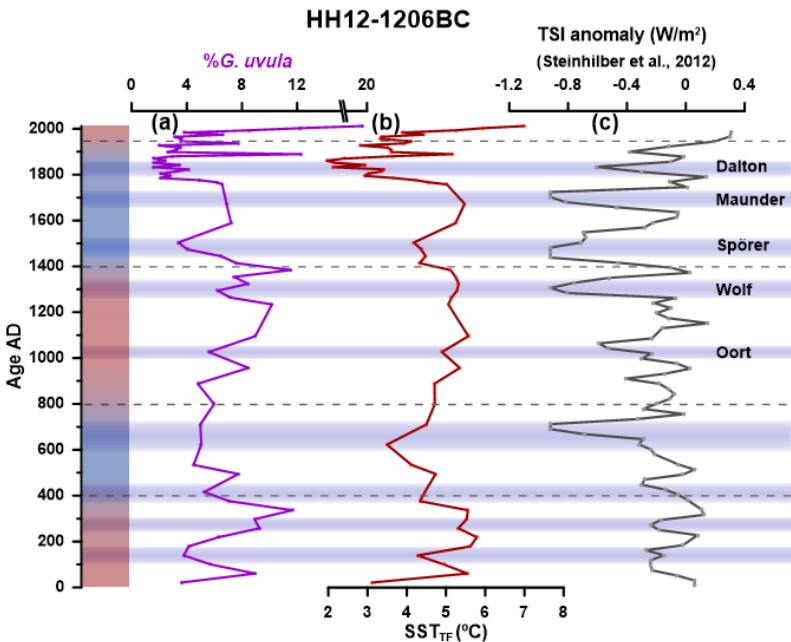

**Figure 9**

Comparison of (a) % of *G. uvula*, (b) SST$_{TF}$ in core HH12-1206BC to (c) TSI anomalies with solar minima (Steinhilber et al., 2012) of the last two millennia. Known solar minima during the last 1000 years; Oort (1010–1050 AD), Wolf (1280–1350 AD), Spörer (1460–1550 AD), Maunder (1645–1715 AD) and Dalton (1790–1820 AD) (Eddy, 1978; Schröder, 2005) are indicated. Black thick lines show 6-point running data in (a) and (b), and 40-point smoothed data in (c). Left panel shows warm (red) and cold (blue) climatic periods during the past 2000 years.