# Peer review of "Planktic foraminifera and structure of surface water masses at the SW Svalbard margin in relation to climate changes during the last 2000 years"

_Climate of the Past, 2018_

## Referee Comment (RC1) · Anonymous Referee #1 · 25 Sep 2018

The paper present novel data on modern and fossil planktic foraminifera in the southwestern Svalbard margin. Planktic foraminifers can be sensitive indicators of the changing Arctic environments. Living planktic foraminifera were collected at different vertical depth-sampling intervals in October 2012 and July 2014 at the coring site. The short (30.5 cm length sampled at 0.5 cm intervals) sediment core was collected in Storfjorden Fan at 1520 m water depth in October 2012 for fossil foraminifers studying. The environmental reconstructions (first of all sea surface temperatures) over the SW Svalbard margin are within the scope of CP. The paper is technically sound. The paper

contains the novel and interesting research of the planktic foraminifera and structure of surface water masses at the SW Svalbard margin in relation to climate changes during the last 2000 years. Conclusions in the paper correspond to the objectives stated in the Introduction. The claims fully supported by the experimental data. The research was carried out at a sufficiently high scientific level. The authors provided sufficient methodological details that the methods could be reproduced. I think the phrase "Trace element ratios of Mg/Ca and Al/Ca..." on the page 7 may be replaced by "Elemental ratios of Mg/Ca and Al/Ca..." There are enough data for providing the age model based on six 14C dates and 210Pb and 137Cs dating as well. The authors clearly indicate their own original contribution into relevant scientific questions. The list of references is quite huge. But I would like to see author's comments about their own sedimentation rates data and climatic inferences in the focus of previous related papers on Svalbard margin (Winkelmann, Knies, 2005; Pathirana et al., 2014, 2015; Vare et al., 2010 and so on). The authors made their paleoceanographical reconstructions based on one short core collected in Storfjorden Fan (SW Svalbard margin). More than 40 short cores were collected west off Svalbard and on the adjacent shelf to the south. I think the results of these studies should be reflected in the Introduction and in the Discussion of the paper. Cited papers: 1. Pathirana, I., Knies, J., Felix, M., Mann, U. (2014) Towards an improved organic carbon budget for the western Barents Sea shelf, Clim. Past, 10, 569-587, https://doi.org/10.5194/cp-10-569-2014. 2. Pathirana I, Knies J, Felix M, Mann U, Ellingsen I (2015) Middle to late Holocene paleoproductivity reconstructions for the western Barents Sea: a model-data comparison, Arktos (2015) 1:20. doi: 10.1007/s41063-015-0002-z 3. Winkelmann D., Knies J. (2005) Recent distribution and accumulation of organic carbon on the continental margin westoff Spitsbergen, Geochem. Geophys. Geosyst., 6, Q09012, doi:10.1029/2005GC000916. 4. Vare LL, Masse G, Belt ST (2010) A biomarker-based reconstruction of sea ice conditions for the Barents Sea in recent centuries. Holocene 20:637–643. doi:10.1177/0959683609355179 The title of the paper reflects the researches results and the contents of the paper. I think the abstract provide a concise

and complete summary. I suppose the presentation is well structured and clear. The language is quite fluent but sometimes probably not precise. Please bear in mind that I am not a native English speaker. The figures and graphs are effective. But I recommend improving Fig. 1. I would like to see the schematic map of Svalbard margin contains the position of short sediment core (probably some short sediment cores from related papers which will be discussed by authors), and surface circulation pattern, bathymetry. The amount and quality of supplementary material is quite appropriate.

---

## Referee Comment (RC2) · Anonymous Referee #2 · 7 Nov 2018

This paper shows severe weaknesses in all parts of the text:

1) The methodology is generally not explained clearly enough, which makes the further interpretation rather unconvincing: - The following sentence (Chapter 2.1, page 5) raises doubts about the good faith of the authors... "The depth-sampling intervals were assigned based on the distribution of water masses recorded by the CTD". This is simply not true. How can one imagine that there is no change in the vertical water stratification between October and July....Of course there is (see Figure 2)! The 0-50m interval is only correct for the October sampling. In other seasons, the authors sampled a mixture of populations living in the mixed layer and below the thermos/haloclines. The further interpretation based on living faunas is therefore severely compromised. - Planktic foraminiferal (PF) faunas are given in number of specimens per cubic meter (Chapter 2.1, page 5). How was the volume of filtered water measured? The WP2 device cannot be equipped with 2 flowmeters, able to record the "in" and "out" fluxes. Thus to my knowledge, it is difficult to accurately estimate the number of ind/m3 with a WP2 sampler. A large error margin has to be taken into account, which was not done at all in the paper. - Chapter 2.2, page 5: what kind of box core was used? Are the authors sure that the water-sediment interface was properly sampled? If the sampler is not completely closed, a washout may occur when the sampler is being retrieved. Consequently, to what extent are the PF counts in the surficial sediment correct? Since there I some uncertainty about the quality of this sample, the core top sample should perhaps be discarded. - Chapter 2.2, page 6: A sentence states "The small (100–125$\mu$m)- and large (150–180$\mu$m)-size shells represent different life stages, the juvenile and adult forms, respectively" I don't agree with this discrimination of different life stages on this basis of very similar test sizes! The size-fractions proposed here are not appropriate. For example, for T.quinqueloba, 125$\mu$m diameter is a "normal" adult size. It is a well-known small species... (e.g. Schiebel and Hemleben, 2005; Husum and Hald, 2012). - Chapter 2.2, page 6: Counting shell fragments is far from trivial. What did the authors count precisely? what sizes of fragments? how to be sure to count only PF fragments?

2) Some basic interpretations are not correct - Chapter 3.2.2: the authors consider fragmentation as an indicator of bad preservation. Which is correct, but the authors focussed only on CaCO3 dissolution to explain fragmentation, writing (page 17) "% fragmentation is low indicting well-saturated with respect to calcium carbonate...". Bad preservation can also be due to transport and/or reworking on the sea floor in areas of active bottom currents. Above all, CaCO3 preservation may be closely linked to early diagenetic processes within the sediment that have nothing to do with the bottom sea characteristics during the deposition! - Chapter 3.2.4, page 15: "the three

CTD casts taken during key-seasons for reproduction of planktic foraminifera at the core site". What are these key-seasons for PF reproduction at the studied site? The authors suggests that PF reproduction occurs 3 times a year, October, April and July. On which data is this free assumption based? What is the real timing of PF reproduction at the core site? To my knowledge, in this area there is absolutely no information available about PF reproduction seasons. - Chapter 3.1.3, page 11: One can read "..G. uvula was found at X... m water depth...., which could indicate that this depth is the calcification depth of the species". Unfortunately, the depth of calcification is generally not where you find most individuals! Calcification of PF starts at the reproduction level, possibly close to the pycnocline (for the studied species; Schiebel and Hemleben, 2005), and ends where PF are largest (i.e. end of calcification), just before the reproduction .... at the same pycnocline level! Calcification depth is still a matter of debate. Calcification depths could have been be discussed here with data of the modern fauna (see my point 3). - Chapter 3.1.3 - on page 12 is written: "This species is capable of living in salinities of 30.5–31, which appears to be the minimum salinity for planktic foraminifera (Boltovskoy and Lena, 1970b)". The authors should have a better look at the available, recent literatures! PF have a high tolerance to salinity changes (e.g., Bijma et al., 1990; Ortiz et al., 1995). They are not directly affected by low salinity (e.g., Fernandez et al., 1991), but rather by parameters that co-vary with salinity (e.g., Ufkes et al., 1998; Retailleau et al, 2009).

3) No isotope measurements were performed on the modern living fauna whereas the authors have good material to do this (plankton tows and Rosette CTD). They should have tested all basic interpretations from the literature with their own data set. For example, - compare 18O and 13C of the living fauna (trapped in the plankton tows) with measurements of the ambient seawater isotopes. This could help to identify the specific calcification depth. - compare the isotopic differences between living species with the observed water masses and subsequent stratification and thus verify the hypotheses presented on page 3 " N. pachyderma and T. quinqueloba ($\Delta\delta$18ONp-Tq) as an indicator of the subsurface-to-near surface Atlantic Water relative inflow and between T. quinqueloba and G. uvula ($\Delta\delta$18OTq-Gu) as an indicator of relative changes in freshening and stratification of the surface waters in the past."

4) Chapter 1.1 Oceanographic setting, gives in 10 lines a very vague and extremely general view of the studied system (without any reference to literature). It is more than necessary to explain here the seasonal and interannual variability of the intensity and location of the modern AF, on the basis of recent oceanographic measurements.

5) In chapter 4.2 Sea surface reconstruction. . ., the authors repeat/summarize their results and interpretation for (only) a single 30cm-long core. No clear comparison with other cores sampled in the vicinity (Eastern North Atlantic, west Svalbard) is presented. But a comparison is made with results from the broadly diversified North Atlantic Ocean domain! It would be very surprising if exactly the same processes influence PF faunas in the eastern Labrador area and along western Svalbard! Such comparison, without any reserve, suggests an ignorance of the functioning of the modern oceans. There is no real discussion, with sentences advocating the "success" of their study because it agrees with others! See in chapter 4.2.1 "The warm surface conditions... are in agreement with other studies from the North Atlantic Ocean Âż; in chapter 4.2.5 "There is a paleoproxy consensus of the progressive warming . . .." Or the worst, in chapter 4.2.2 "Overall, taking into account differences in sedimentation rates, dating control, and marine reservoir age corrections, a warming . . .in the Storfjorden Fan can be considered as a widespread phenomenon"

––––––––––––––––––––––––––

---

## Author Comment (AC1) · 4 Dec 2018

**Detailed response to the Reviewers' comments**

We thank the Anonymous Reviewer #1 and Anonymous Reviewer #2 for the time devoted on reading and commenting the manuscript and the important suggestions for improvements. We address each comment given by the reviewers below.

All our additions and changes made in the manuscript are listed by chapters, pages and lines in the reply letter and marked in yellow in revised manuscript. In addition, changes (in italic) are pasted in below each reply to the reviewers (in most cases).

\*reviewer comment are in black and answers to those comments are in blue

**Anonymous Reviewer #1**

The paper present novel data on modern and fossil planktic foraminifera in the southwestern Svalbard margin. Planktic foraminifers can be sensitive indicators of the changing Arctic environments. Living planktic foraminifera were collected at different vertical depth-sampling intervals in October 2012 and July 2014 at the coring site. The short (30.5 cm length sampled at 0.5 cm intervals) sediment core was collected in Storfjorden Fan at 1520 m water depth in October 2012 for fossil foraminifers studying. The environmental reconstructions (first of all sea surface temperatures) over the SW Svalbard margin are within the scope of CP. The paper is technically sound. The paper contains the novel and interesting research of the planktic foraminifera and structure of surface water masses at the SW Svalbard margin in relation to climate changes during the last 2000 years. Conclusions in the paper correspond to the objectives stated in the Introduction. The claims fully supported by the experimental data. The research was carried out at a sufficiently high scientific level. The authors provided sufficient methodological details that the methods could be reproduced.

I think the phrase "Trace element ratios of Mg/Ca and Al/Ca" on the page 7 may be replaced by "Elemental ratios of Mg/Ca and Al/Ca"

We thank Reviewer#1 for the kind and positive comments. We have replaced the phrase "Trace element ratios of Mg/Ca and Al/Ca" by "*Elemental ratios of Mg/Ca and Al/Ca*" as requested, p. 8, line 7.

\_\_\_\_\_

There are enough data for providing the age model based on six 14C dates and 210Pb and 137Cs dating as well. The authors clearly indicate their own original contribution into relevant scientific questions. The list of references is quite huge. But I would like to see author's comments about their own sedimentation rates data and climatic inferences in the focus of previous related papers on Svalbard margin (Winkelmann, Knies, 2005; Pathirana et al., 2014, 2015; Vare et al., 2010 and so on).

We thank the Reviewer#1 for this suggestion and indication of relevant publications. Based on the pointed out studies we added a comment on our sedimentation rates data with references in chapter Results, 3.2.1, p. 12, lines 28-31:

"These rates are similar to sedimentation rate reported for the Barents Sea which range from 0.06–0.09 cm yr-1 to 0.10–0.14 cm yr-1 dependent on whether the upper mixed layer is included in the calculations or not, respectively (Maiti et al., 2010; Vare et al., 2010; Pathirana et al., 2014; 2015 and references therein)."

The authors made their paleoceanographical reconstructions based on one short core collected in Storfjorden Fan (SW Svalbard margin). More than 40 short cores were collected west off Svalbard and on the adjacent shelf to the south. I think the results of these studies should be reflected in the Introduction and in the Discussion of the paper. Cited papers:

1. Pathirana, I., Knies, J., Felix, M., Mann, U. (2014) Towards an improved organic carbon budget for the western Barents Sea shelf, Clim. Past, 10, 569-587, https://doi.org/10.5194/cp-10-569-2014.

2. Pathirana I, Knies J, Felix M, Mann U, Ellingsen I (2015) Middle to late Holocene paleoproductivity reconstructions for the western Barents Sea: a model-data comparison, Arktos (2015) 1:20. doi: 10.1007/s41063-015-0002-z

3. Winkelmann D., Knies J. (2005) Recent distribution and accumulation of organic carbon on the continental margin west off Spitsbergen, Geochem. Geophys. Geosyst., 6, Q09012, doi:10.1029/2005GC000916.

4. Vare LL, Masse G, Belt ST (2010) A biomarker-based reconstruction of sea ice conditions for the Barents Sea in recent centuries. Holocene 20:637–643. doi:10.1177/0959683609355179

We have followed the reviewer's suggestion and included the results of the suggested publications in the introduction (p. 2, line 15 and 25; p. 4, line 12-15), discussion section and the new chapter 4.3 Comparison to other studies (p. 21-23).

The title of the paper reflects the researches results and the contents of the paper. I think the abstract provide a concise and complete summary. I suppose the presentation is well structured and clear. The language is quite fluent but sometimes probably not precise. Please bear in mind that I am not a native English speaker.

We have shortened the manuscript and re-formulated some parts of the text in a more precise way. We also corrected the English language.

The figures and graphs are effective.

But I recommend improving Fig. 1. I would like to see the schematic map of Svalbard margin contains the position of short sediment core (probably some short sediment cores from related papers which will be discussed by authors), and surface circulation pattern, bathymetry. The amount and quality of supplementary material is quite appropriate

We have included a schematic map of the Svalbard margin showing the surface circulation pattern, bathymetry and the position of the sediment cores discussed in the manuscript as recommended by the Reviewer, see below.

**Anonymous Reviewer #2**

1) The methodology is generally not explained clearly enough, which makes the further interpretation rather unconvincing:

The following sentence (Chapter 2.1, page 5) raises doubts about the good faith of the authors. "The depth-sampling intervals were assigned based on the distribution of water masses recorded by the CTD". This is simply not true. How can one imagine that there is no change in the vertical water stratification between October and July. Of course there is (see Figure 2)! The 0-50m interval is only correct for the October sampling. In other seasons, the authors sampled a mixture of populations living in the mixed layer and below the thermos/haloclines. The further interpretation based on living faunas is therefore severely compromised.

We acknowledge our mistake in the statement (Chapter 2.1, p. 5, line 16-17) in the submitted version of our manuscript. As Reviewer #2 is pointing out this information is only valid for the October 2012, the time of our core sampling, we have therefore made the necessary changes to the manuscript.

Further, we agree that in other seasons, in 0-50 m water depth we collected a mixture of populations (NB we always collect all populations in these depth-intervals of the water column, never a single population) living in the mixed layer and below the thermo/halocline. As we compare closely to the living planktic foraminifera fauna data from other studies/literature in this region (chapter 3.1.3 Modern distribution and ecology of living high-latitude planktic foraminifera), we think that our interpretation is valid and comparable to other studies. Nevertheless, in order to clarify the issue pointed out by Reviewer #2, we removed the sentence and added the information to Chapter: 3.1.3, new p. 11, lines 6-10: *"As our plankton tow data are collected in two different years, at two different seasons and different depth intervals (see Methods), we cannot examine possible species-specific living*

depths. We therefore support our data with published results to establish a record of the

**modern vertical distribution of planktic foraminiferal species in the surface water column of the study area."**
* * *
Planktic foraminiferal (PF) faunas are given in number of specimens per cubic meter (Chapter 2.1, page 5). How was the volume of filtered water measured? The WP2 device cannot be equipped with 2 flowmeters, able to record the "in" and "out" fluxes. Thus to my knowledge, it is difficult to accurately estimate the number of ind/m3 with a WP2 sampler. A large error margin has to be taken into account, which was not done at all in the paper.

There are two ways to estimate the volume of filtered water: (1) by using a flowmeter and, if a flowmeter cannot be used, (2) by multiplying the net aperture area by depth towed. As the Reviewer #2 correctly points out the WP2 plankton sampler cannot be equipped with a flowmeter, so we use method (2), the calculation to estimate the volume of water that passed through the net (the volume of a cylinder with the length represented by the depth of the tow). In both methods, the accuracy has some errors:

Flowmeters don't work well for vertical tows, because they record the current going down as well and since a perfectly vertical tow recovery is extremely rare (or impossible in moving sea), one have to note the angle the haul line makes with the surface of the water and calculate the hypotenuse to get the true depth.

Method (2) can introduce some uncertainties because mesh-sizes below  $200\mu m$  can get clogged easily during spring blooms or when gelatinous zooplankton are abundant. As the net becomes progressively clogged as it ascend, it filters less water near the surface than at depth. Also, if one tow up to quickly (faster than 1-2m per second, you may create a pressure wave in front of the rapidly moving net that pushes aside some zooplankton.

During our sampling, we did not record any zooplankton/ algae overabundance and the vertical towing speed was ~  $0.5 \text{ m s}^{-1}$  (Chapter 2.1, line 17-18) which assured proper and complete recovery of planktic foraminifera (zooplankton).

We have now added how the volume and the number of  $ind/m^3$  were calculated to Chapter 2.1, p. 5, lines 16-19:

"Vertical towing speed was ~0.5 m s-1 and the volume of filtered water was measured by multiplying net aperture area by depth towed following the general formula:  $V = \pi r^2 \times L$ , where V = volume (m3) of seawater,  $\pi = 3.1415$ ,  $r^2 =$  radius of net opening squared (m2), L = length, distance net was towed (m)."

Chapter 2.2, page 5: what kind of box core was used?

Are the authors sure that the water-sediment interface was properly sampled? If the sampler is not completely closed, a washout may occur when the sampler is being retrieved. Consequently, to what extent are the PF counts in the surficial sediment correct? Since there I some uncertainty about the quality of this sample, the core top sample should perhaps be discarded.

Technical data of the box core used are: Total height: 2.5 meters; Total length after release is 2.5 meter + 3.3 meter wire = 5.8 meter; The box core is 5-sided where: Shortest wall: 1.95 meter, Longest wall: 2.25 meters; Weight: 1100 kg; Volume of test box: 125 liters; Box dimensions: 50x50x50 cm.

We agree with Reviewer #2 that if the sampler is not completely closed a washout may occur when the sampler is being retrieved. We are well aware of the potential problem, hence in the

rare cases, if the box corer is disturbed, we always consider the sampling as failure and make a new attempt of coring. Furthermore, we are certain that the water-sediment interface was properly sampled in our case. As documentation we include photos of the surface sediment of the boxcore HH12-1206BC after retrieval and removal of seawater (Fig. 1a,b). Moreover, as stated in the manuscript (Results, Chapter 3.2.1, line 13-15) profiles analyzed for activity of 210Pb and 137Cs clearly show that 3 cm thick surface layer has the highest and near-uniform activities indicating a surface mixed layer, thereby confirming the recovery of the upper surface sediments of the core.

---

## Author Comment (AC2) · 4 Dec 2018

We kindly thank the Anonymous Reviewer #2 for their insightful comments that helped us to improve the manuscript. We attached our detailed response to the Reviewers' comments as a pdf.

Please also note the supplement to this comment: https://www.clim-past-discuss.net/cp-2018-93/cp-2018-93-AC2-supplement.pdf